# Towards Sharper Generalization Bounds for Structured Prediction

**Shaojie Li**[1,2]     **Yong Liu**[1,2,*]
[1]Gaoling School of Artificial Intelligence, Renmin University of China, Beijing, China
[2]Beijing Key Laboratory of Big Data Management and Analysis Methods, Beijing, China
`2020000277@ruc.edu.cn, liuyonggsai@ruc.edu.cn`

## Abstract

In this paper, we investigate the generalization performance of structured prediction learning and obtain state-of-the-art generalization bounds. Our analysis is based on factor graph decomposition of structured prediction algorithms, and we present novel margin guarantees from three different perspectives: Lipschitz continuity, smoothness, and space capacity condition. In the Lipschitz continuity scenario, we improve the square-root dependency on the label set cardinality of existing bounds to a logarithmic dependence. In the smoothness scenario, we provide generalization bounds that are not only a logarithmic dependency on the label set cardinality but a faster convergence rate of order $\mathcal{O}(\frac{1}{n})$ on the sample size $n$. In the space capacity scenario, we obtain bounds that do not depend on the label set cardinality and have faster convergence rates than $\mathcal{O}(\frac{1}{\sqrt{n}})$. In each scenario, applications are provided to suggest that these conditions are easy to be satisfied.

## 1   Introduction

Structured prediction [18, 12, 17] covers a wide range of machine learning fields, such as computer vision, natural language processing, and computational biology. Several examples of structured prediction problems include part-of-speech tagging, dependency parsing, named entity recognition and machine translation in natural language understanding, image segmentation and objection recognition in computer vision, and protein folding in computational biology. An essential property of structured prediction is that the output space admits some structure, such as strings, graphs, trees, or sequences [18, 12, 13]. Meanwhile, another property common to the above tasks is that the natural loss function in each case admits a decomposition along with the output substructures. The complex output structure and corresponding loss function make structured prediction different from the widely studied binary or multi-class classification problems.

The design of structured prediction algorithms have been thrived for many years, including conditional random fields [30], structured support vector machines [61], kernel-regression algorithm [19], search-based structured prediction [20], maximum-margin markov network [60], image segmentation [43], part-of-speech-tagging [27], and machine translation [68]. Compared to the prosperity and development of structured prediction algorithm designing, the theoretical analysis of structured prediction appears to be not sufficiently well-documented [13], especially the studying on sharper generalization bounds. However, it is known that the theoretical study of structured prediction algorithms is essential [18, 12, 13, 1].

Several theoretical studies of structured prediction consider a specific algorithm and a simple loss such as Hamming loss [60, 16, 14]. Recently, increasing work has considered the analysis of general losses. These works can be roughly cast into four categories: PAC-Bayesian approach, factor graph

---

*Corresponding Author.

decomposition approach, implicit embedding approach, and stability approach. [22, 45, 26, 4] provide PAC-Bayesian guarantees for arbitrary losses through the analysis of randomized algorithms using count-based hypotheses. [12, 13, 53, 52, 8, 11, 58, 7] use implicit loss embedding to construct a connection between discrete output of structured prediction and regression, to investigate the generalization performance of structured prediction. [18, 3, 51] present generalization bounds for general losses based on factor graph decomposition approach, which is proposed in the seminal work [18]. [41, 42, 40] use the stability tool to investigate the generalization bounds and provide interesting theoretical insights that structure prediction is possible to generalize from a few large examples, potentially even just one. A special case of structured prediction problem worth mentioning is multi-class classification [34, 33, 36, 55, 50, 65, 48, 49, 44, 31], whose scoring function can be seen as a factor graph with both factor node size and substructure size equal to one.

Although the aforementioned works have provided generalization bounds from different perspectives, there are still some problems urgently to be solved in structured prediction.

(1.) Existing generalization bounds mostly have a slow convergence rate with respect to (w.r.t.) the sample size $n$. Specifically, in PAC-Bayesian approach, [45, 26, 4, 22] provide the generalization bounds of order $\mathcal{O}(\frac{1}{\sqrt{n}})$. In implicit embedding approach, [12, 13, 52, 11, 58, 7] provide the convergence rate of order $\mathcal{O}(\frac{1}{n^{1/4}})$, and [53] of order $\mathcal{O}(\frac{1}{\sqrt{n}})$. In the factor graph decomposition approach, [18, 51] present the generalization upper bounds of order $\mathcal{O}(\frac{1}{\sqrt{n}})$. Also, in stability approach, [41, 42, 40] show the $\mathcal{O}(\frac{1}{\sqrt{n}})$ order bounds. Therefore, this raises a question: can the convergence rate of structured prediction achieves faster order $\mathcal{O}(\frac{1}{n})$?

(2.) In structured prediction, the number of possible labels is potentially infinite [18]. Thus it is important to obtain upper bounds that have a lower order dependency on the cardinality of possible labels. The factor graph decomposition approach can provide the explicit dependency on the properties of the factor graph and help us to explicitly show the dependency on the number of possible labels, which sheds new light on the role played by the graph in generalization [18]. Therefore, our analysis is based on the factor graph decomposition [18, 3, 51]. However, [3] focuses on the lower bound, [51] studies the specific surrogate loss, and the upper bounds in [18] show a square-root dependency on the cardinality of possible labels. If we consider the vast number of possible labels, the results in [18] may be divergent and can not explain the good performance of structured prediction algorithms in practice. Thus, can the upper bound of structured prediction presents a lower order dependency on the cardinality of possible labels or even no dependency?

(3.) Although some work in multi-class classification [36, 55, 65], a special structured prediction problem, have shown the faster convergence rate of order $\mathcal{O}(\frac{1}{n})$ w.r.t. the sample size $n$, it is unknown in more difficult and complex structured prediction problems. Additionally, we naturally want to know whether the generalization bounds of structured prediction can combine (1) with (2) to show faster convergence rates simultaneously?

This paper intends to answer the three interesting questions. We consider the general loss and present novel margin-based theoretical analysis for structured prediction from three perspectives: Lipschitz continuity, smoothness, and space capacity condition. We first assume the loss function used in structured prediction is Lipschitz continuous and try to answer question (2). Under this condition, we improve the generalization bounds of structured prediction from a square-root dependency to a logarithmic dependency on the label set cardinality, but a slow convergence rate of order $\mathcal{O}(\frac{1}{\sqrt{n}})$ w.r.t. the sample size $n$. Then, we assume the loss is smooth and intend to answer questions (1) and (3). Under this assumption, we obtain sharper bounds that not only have the faster convergence rate of order $\mathcal{O}(\frac{1}{n})$ on the sample size $n$ but also have the logarithmic dependency on the label set cardinality. Furthermore, we consider the space capacity-dependent assumptions: logarithmic covering number and polynomial covering number assumptions, which are commonly used in learning theory [32, 69, 56]. This setting attempts to answer questions (1-3). Under this condition, we show that the bound can present no dependency on the label set cardinality, simultaneously with faster convergence rates than $\mathcal{O}(\frac{1}{\sqrt{n}})$. In the above three perspectives, we all provide applications to suggest that these conditions are easy to be satisfied in practice.

This paper is organized as follows. In Section 2, we introduce the notations and definitions relevant to our discussion. In Section 3, we present the main results, deriving a series of new learning guarantees for structured prediction. Section 4 concludes this paper.

## 2 Preliminaries

**Notations.** Let $P$ be a probability measure defined on a sample space $\mathcal{X} \times \mathcal{Y}$ with $\mathcal{X} \in \mathbb{R}^d$ being the input space and $\mathcal{Y}$ being the output space. A key aspect of structured prediction is that the output space may be sequences, graphs, images, or lists [18]. We thus assume that $\mathcal{Y}$ can be decomposed into $l$ substructures: $\mathcal{Y} = \mathcal{Y}_1 \times \cdots \times \mathcal{Y}_l$, where $\mathcal{Y}_i$ is the set of possible labels that can be assigned to substructure $i$. Take the simple webpage collective classification task for example [59], each $\mathcal{Y}_i$ is a webpage label and $\mathcal{Y}$ is a joint label for an entire website. If we assume each $\mathcal{Y}_i \in \{0, 1\}$, the number of possible labels to $\mathcal{Y}$ is exponential in the number of substructures $l$, i.e., $|\mathcal{Y}| = 2^l$.

**Factor graphs and scoring function.** In structured prediction, we aim to learn a scoring function $f = \mathcal{X} \times \mathcal{Y} \to \mathbb{R}$. Let $\mathcal{F}$ be a family of the scoring function. For each $f \in \mathcal{F}$, we define the predictor $\hat{f}$ as: for each $x \in \mathcal{X}$, $\hat{f}(x) = \arg\max_{y \in \mathcal{Y}} f(x, y)$. Moreover, we assume that the scoring function $f$ can be decomposed as a sum [61, 60, 30, 18], which is standard in structured prediction and can be formulated via the notation of factor graph [18]. A factor graph $G$ is a bipartite graph, and is represented as a tuple $G = (V, H, E)$, where $V = \{1, ..., l\}$ is a set of variables nodes, $H$ is a set of factor nodes, and $E$ is a set of undirected edges between a variable node and a factor node. Let $\mathcal{N}(h)$ be the set of variable nodes connected to the factor $h$ by an edge and $\mathcal{Y}_h$ be the substructure set cross-product $\mathcal{Y}_h = \prod_{k \in \mathcal{N}(h)} \mathcal{Y}_k$. Based on the above notations, the scoring function $f(x, y)$ can be decomposed as a sum of functions $f_h$ [18], with an element $x \in \mathcal{X}$ and an element $y_h \in \mathcal{Y}_h$ as arguments:

$$f(x, y) = \sum_{h \in H} f_h(x, y_h).$$

In this paper, we consider the more generally setting: for each example $(x_i, y_i)$, the corresponding factor graph may be different, that is, $G(x_i, y_i) = G_i = ([l_i], H_i, E_i)$. A special case of this setting is that, for example, when the size $l_i$ of each example is allowed to vary and where the number of possible labels $|\mathcal{Y}|$ is potentially infinite. Figure 1 shows different examples of factor graphs.

**Learning.** In order to measure the success of a prediction, we use a loss function $L : \mathcal{Y} \times \mathcal{Y} \to \mathbb{R}_+$ to measure the dissimilarity of two elements of the output space $\mathcal{Y}$. A distinguishing property of structured prediction is that the natural loss function admits a decomposition along the output substructures [18], such as the Hamming loss, defined as: $L(y, y') = \frac{1}{l} \sum_{k=1}^{l} \mathbb{I}_{y_k \neq y'_k}$ for two outputs $y, y' \in \mathcal{Y}$, with $y = (y_1, ..., y_l)$ and $y = (y'_1, ..., y'_l)$, or edit-distance loss in natural language and speech processing applications, or other losses [18, 10]. Given a training set $S = \{(x_1, y_1), ..., (x_n, y_n)\}$ of size $n$ being independently drawn from the underlying distribution $P$, our goal is to learn a hypothesis $f \in \mathcal{F}$ with good generalization performance from the sample $S$ by optimizing the loss $L$. The performance is measured by expected risk, defined as

$$R(f) := \mathbb{E}_{(x,y) \sim P}[L(\hat{f}(x), y)].$$

However, the above mentioned loss function $f \to L(\hat{f}(x), y)$ is typically the $0 - 1$ loss of $f$, which is hard to handle in machine learning. Therefore, one usually consider surrogate losses [12, 13, 18].

**Margin and loss function class.** We now introduce the definition of standard margin and the surrogate loss. The margin of any scoring function $f$ for an example $(x, y)$ is defined as:

$$\rho_f(x, y) = f(x, y) - \max_{y' \neq y} f(x, y').$$

We consider the scoring function $f$ of the form $f(x, y) = \langle w, \Psi(x, y) \rangle$, where $\Psi$ is a feature mapping from $\mathcal{X} \times \mathcal{Y}$ to $\mathbb{R}^N$ such that $\Psi(x, y) = \sum_{h \in H} \Psi_h(x, y_h)$ due to the decomposition of $f$ [18]. And we define the following scoring function space:

$$\mathcal{F} = \{x \mapsto \langle w, \Psi(x, y) \rangle : w \in \mathbb{R}^N, \|w\|_p \leq \Lambda_p\}, \tag{1}$$

where $\|w\|_p = (\sum_{i=1}^{N} |w_i|^p)^{\frac{1}{p}}$. Then, the corresponding surrogate loss function class is defined as:

$$\mathcal{L}_\rho = \{\ell_\rho(x, y, f) := \ell(\rho_f(x, y)) : f \in \mathcal{F}\},$$

where $\ell_\rho$ is the surrogate loss of $L$. We assume $\ell_\rho$ is bounded by $M$, which implies that $0 \leq L(\hat{f}(x), y) = L(\hat{f}(x), y)_{\rho_f(x,y) \leq 0} \leq \ell(\rho_f(x, y)) \leq M$ since $L$ is positive, the cost will only occur

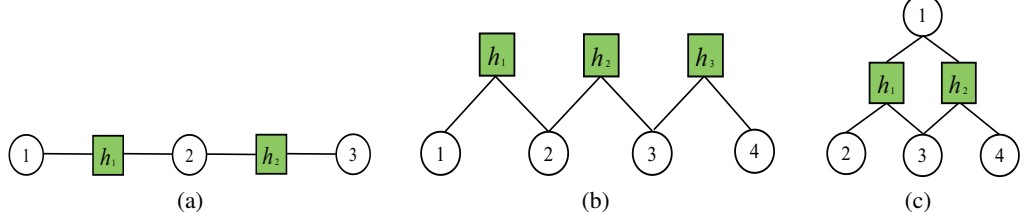

Figure 1: Example of factor graphs. (a) represents an example of pairwise Markov network decomposition: $f(x, y) = f_{h_1}(x, y_1, y_2) + f_{h_2}(x, y_2, y_3)$; (b) represents an example of factor graph that having a sequence structure: $f(x, y) = f_{h_1}(x, y_1, y_2) + f_{h_2}(x, y_2, y_3) + f_{h_3}(x, y_3, y_4)$; (c) represents a tree-structured factor graph.

when the margin function value is negative and the surrogate loss $\ell$ is larger than $L$. Additionally, we define the empirical risk of any scoring function $f$ as

$$\hat{R}(\ell(\rho_f)) := \frac{1}{n} \sum_{i=1}^{n} \ell_\rho(x_i, y_i, f),$$

and the expected risk as

$$R(\ell(\rho_f)) := \mathbb{E}_{(x,y) \sim P}[\ell_\rho(x, y, f)].$$

Clearly, we have $R(f) \leq R(\ell(\rho_f))$.

## 3 Main Results

This section provides sharper generalization bounds for structured prediction from three perspectives: Lipschitz continuity, smoothness, and space capacity. Finally, we obtain sharper bounds of order $\mathcal{O}(\frac{1}{n})$ w.r.t. the sample size $n$ and showing a logarithmic dependency, or even no dependence, on the label set cardinality. In each case, we provide applications to suggest that these conditions are easy to be satisfied.

### 3.1 Lipschitz Continuity Dependent Case

**Assumption 1** *Assume that $\ell$ is $\mu$-Lipschitz continuous, that is*

$$|\ell(t) - \ell(s)| \leq \mu|t - s|.$$

Assumption 1 is a pretty mild assumption. The hinge loss $\ell(\rho_f) = (1 - \rho_f)_+$ and the margin loss $\ell(\rho_f) = 1_{\rho_f \leq 0} + (1 - \rho_f \rho^{-1})1_{0 < \rho_f \leq \rho}$ satisfy Assumption 1 [33]. Moreover, the loss of truncated margin functions in [25] satisfies Assumption 1. The additive and the multiplicative margin loss provided by [18], which covers many existing structured prediction algorithms, satisfy Assumption 1 (see Corollary 1 for details).

**Theorem 1** *Under Assumption 1, for any $\delta > 0$, with probability $1 - \delta$ over the sample $S$ of size $n$, the following holds for all $f \in \mathcal{F}$,*

$$R(f) \leq R(\ell(\rho_f)) \leq \hat{R}(\ell(\rho_f)) + \mathcal{O}\left(\frac{\mu s \ln n}{\sqrt{n}}\left(\log^{\frac{1}{2}}(n^2) + \log^{\frac{1}{2}}(sk)\right) + \sqrt{\frac{\log 1/\delta}{n}}\right),$$

*where $s = \max_{i \in [n]} |H_i|$, $k = \max_{i \in [n]} |H_i| d_i$, and where $d_i = \max_{h \in |H_i|} |\mathcal{Y}_h|$.*

**Remark 1 [Sketch of Theorem 1.]** Since our task is to prove bounds with better rates, that is, which decrease fast with $n \to \infty$, thus for brevity, we defer the explicit results of all theorems to the Appendix. In Theorem 1, we focus on terms in the numerator. From Theorem 1, one can see that our generalization bound has a linear dependency on the maximum factor size of a given sample set, that is $s = \max_{i \in [n]} |H_i|$. The terms of order $\mathcal{O}(\log(\cdot))$ is small enough and typically can be ignored. For terms $\log^{\frac{1}{2}}(n^2)$ and $\log^{\frac{1}{2}}(sk)$, if we consider the case $n^2 \geq sk$, our result suggests that

in the generalization error, the influence of sample size $n$ is larger than the label set cardinality, which implies that if the sample size $n$ is large enough, hyper-parameter $n$ will dominate the generalization performance of structured prediction. While for the label set cardinality, our bound has a logarithmic dependency, that is $\log(d_i)$ (Since the factor size $s$ is typically far smaller than label size $d_i$ [18], we just consider the label set size $d_i$ in the term $\log^{\frac{1}{2}}(sk)$). We now compare our bound with the related work in the factor graph decomposition [18, 51]. When the function class $\mathcal{F}$ is assigned with $p = 1$ or $p = 2$, the generalization bounds in [18] are $R(\ell^{add}(\rho_f)) \leq \hat{R}(\ell^{add}(\rho_f)) + \mathcal{O}(\frac{s\sqrt{d_i}}{\rho\sqrt{n}} + \sqrt{\frac{\log 1/\delta}{n}})$ and $R(\ell^{mult}(\rho_f)) \leq \hat{R}(\ell^{mult}(\rho_f)) + \mathcal{O}(\frac{Ms\sqrt{d_i}}{\rho\sqrt{n}} + \sqrt{\frac{\log 1/\delta}{n}})$ ($\ell^{add}$ and $\ell^{mult}$ are defined in Application 1). Thus the generalization bounds in [18] have a linear dependency on $s$ but a square-root dependency on $d_i$, i.e., $\sqrt{d_i}$. The term $\sqrt{d_i}$ may be extremely large since it is the largest label set cardinality of the factor in $H_i$. If we consider the extremely large number of possible labels, the bounds in [18] may be divergent. Thus, by comparison, our generalization bound has a significant improvement. [51] studies the specific surrogate loss $\ell^{add}$, and they also provide a logarithmic dependency on the label set cardinality. Besides, by considering stationary $\beta$-mixing stochastic process [47], they also provide interesting generalization analysis for learning with weakly dependent data. Unfortunately, their generalization bounds and proof techniques are limited to this specific loss. Compared with their results, our bound is applicable for any Lipschitz continuity losses. In other words, so long as the surrogate loss is Lipschitz continuous w.r.t. the margin function, our bound is applicable. This assumption is pretty mild, as we have discussed below Assumption 1. The proof of Theorem 1 is provided in Appendix A. Additionally, the Lipschitz continuity constant $\mu$ is typically small. We present two applications for examples in the following.

**Application 1.** [18] provides two margin loss, the additive and the multiplicative margin losses, that can be used to guarantee many existing structured prediction algorithms, defined as:

$$\ell^{add}(\rho_f(x, y)) = \Phi^* \left( \max_{y' \neq y} L(y', y) - \frac{1}{\rho} [f(x, y) - f(x, y')] \right),$$

$$\ell^{mult}(\rho_f(x, y)) = \Phi^* \left( \max_{y' \neq y} L(y', y) \left( 1 - \frac{1}{\rho} [f(x, y) - f(x, y')] \right) \right),$$

where $\Phi^*(r) = \min\left(\max_{y, y'} L(y, y'), \max(0, r)\right)$ for any $y, y' \in \mathcal{Y}$. For the additive and the multiplicative margin losses, we show that they are $\frac{1}{\rho}$ and $\frac{M}{\rho}$ Lipschitz continuous w.r.t. the margin function $\rho_f$ respectively in Appendix B, thus we have Corollary 1.

**Corollary 1** *Fix $\rho > 0$. For any $\delta > 0$, with probability $1 - \delta$ over the sample $S$ of size $n$, the following holds for all $f \in \mathcal{F}$,*

$$R(f) \leq R(\ell^{add}(\rho_f)) \leq \hat{R}(\ell^{add}(\rho_f)) + \mathcal{O}\left( \frac{s \ln n}{\rho\sqrt{n}} \left( \log^{\frac{1}{2}}(n^2) + \log^{\frac{1}{2}}(sk) \right) + \sqrt{\frac{\log 1/\delta}{n}} \right),$$

$$R(f) \leq R(\ell^{mult}(\rho_f)) \leq \hat{R}(\ell^{mult}(\rho_f)) + \mathcal{O}\left( \frac{Ms \ln n}{\rho\sqrt{n}} \left( \log^{\frac{1}{2}}(n^2) + \log^{\frac{1}{2}}(sk) \right) + \sqrt{\frac{\log 1/\delta}{n}} \right),$$

*where $s = \max_{i \in [n]} |H_i|$, $k = \max_{i \in [n]} |H_i| d_i$, and where $d_i = \max_{h \in |H_i|} |\mathcal{Y}_h|$.*

**Application 2.** In many structured prediction applications, such as natural language processing and computer vision, people may wish to take advantage of very rich features. However, using a rich family of hypotheses can lead to overfitting. In this application, we further consider to derive learning guarantees for ensembles of structured prediction rules that explicitly account for the differing complexities between families, called Voted Risk Minimization (VRM) [15, 18]. Assume that we are given $p$ families $F_1, ..., F_p$ of functions mapping from $\mathcal{X} \times \mathcal{Y}$ to $\mathbb{R}$. Define the ensemble family $\mathcal{G} = conv(\cup_{i=1}^p F_i)$, that is the family of functions $g$ of the form $g = \sum_{t=1}^T \alpha_t f_t$, where $\alpha = (\alpha_1, ..., \alpha_T)$ is in the simplex $\Delta$ and where, for each $t \in [1, T]$, $f_t$ is in $F_{i_t}$ for some $i_t \in [1, p]$. Consider the following loss function:

$$\ell_\tau^{add}(\rho_g(x, y)) = \Phi^* \left( \max_{y' \neq y} L(y', y) + \tau - \frac{1}{\rho} [g(x, y) - g(x, y')] \right),$$

$$\ell_\tau^{mult}(\rho_g(x, y)) = \Phi^* \left( \max_{y' \neq y} L(y', y) \left( 1 + \tau - \frac{1}{\rho} [g(x, y) - g(x, y')] \right) \right),$$

where $\tau$ can be seen as a margin term that acts in conjunction with $\rho$. We assume these families $F_1, ..., F_p$ have differing complexities. This setting is the same as [18], and under this setting we have Corollary 2 for VRM.

**Corollary 2** *Fix $\rho > 0$. For any $\delta > 0$, with probability $1 - \delta$ over the sample $S$ of size $n$, the following holds for all $g \in \mathcal{G}$,*

$$R(g) \leq \hat{R}(\ell_1^{add}(\rho_g)) + \mathcal{O}\left(\frac{s \ln n}{\rho \sqrt{n}}\left(\log^{\frac{1}{2}}(n^2) + \log^{\frac{1}{2}}(sk)\right) + \sqrt{C(n, p, \rho, |\mathcal{Y}|, \delta)}\right),$$

$$R(g) \leq \hat{R}(\ell_1^{mult}(\rho_g)) + \mathcal{O}\left(\frac{Ms \ln n}{\rho \sqrt{n}}\left(\log^{\frac{1}{2}}(n^2) + \log^{\frac{1}{2}}(sk)\right) + \sqrt{C(n, p, \rho, |\mathcal{Y}|, \delta)}\right),$$

*where $C(n, p, \rho, |\mathcal{Y}|, \delta) = \lceil\frac{1}{\rho^2}\log(\frac{|\mathcal{Y}|^2\rho^2 n}{4\log p})\rceil\frac{\log p}{n} + \frac{\log 2/\delta}{n}$, $s = \max_{i \in [n]} |H_i|$, $k = \max_{i \in [n]} |H_i| d_i$, and where $d_i = \max_{h \in |H_i|} |\mathcal{Y}_h|$.*

**Remark 2** [**Sketch of Applications 1 and 2.**] Corollary 1 shows that, under the same assumption, our generalization bounds improve results in [18] from a square-root dependency $\sqrt{d_i}$ to a logarithmic dependency $\log(d_i)$. For VRM in Application 2, [18] shows that $R(g) \leq \hat{R}(\ell_1^{add}(\rho_g)) + \mathcal{O}(\frac{1}{\rho\sqrt{n}}\sum_{t=1}^T \alpha_t \Re^G(F_{k_t}) + \sqrt{C(n, p, \rho, |\mathcal{Y}|, \delta)})$ and $R(g) \leq \hat{R}(\ell_1^{mult}(\rho_g)) + \mathcal{O}(\frac{M}{\rho\sqrt{n}}\sum_{t=1}^T \alpha_t \Re^G(F_{k_t}) + \sqrt{C(n, p, \rho, |\mathcal{Y}|, \delta)})$. Thus, the generalization bounds of VRM in [18] are dominated by term $\sum_{t=1}^T \alpha_t \Re^G(F_{k_t})$, where $\Re^G(F_{k_t})$ is factor graph Rademacher complexity of the function class $F_{k_t}$ (please refer to their paper for definition). The explicit dependency of their bounds on parameter vector $\alpha$ reveals that learning even with highly complex hypothesis sets could be possible so long as the complexity term. However, Corollary 2 shows the explicit bounds rather than the $\Re^G(F_{k_t})$ term and suggests that the generalization bounds of VRM do not depend on the parameter vector $\alpha$, which implies that a huge number of complex families are allowed. Therefore, it is feasible to use rich families in conjunction with highly complex families in structured prediction. Furthermore, in Theorem 2 of [18], the term $\Re^G(F_{k_t})$ is of order $\frac{\sqrt{\max_{i \in [n]} |H_i|^2 d_i}}{\sqrt{n}}$, where $d_i = \max_{h \in |H_i|} |\mathcal{Y}_h|$, which presents a square-root dependency on the label size, that is $\sqrt{d_i}$. By comparison, our bounds are tighter. Therefore, the generalization bounds in Corollary 2 provide further insights into the learning guarantees of VRM. The complete proof of Corollary 1 and Corollary 2 is provided in Appendix B and C, respectively.

**Remark 3** We now consider two examples in practical situations and further demonstrate our results.

(1.) Consider the pairwise Markov networks with a fixed number of substructures $l$ studied by [60, 18], the corresponding factor graph in our paper has $l$ nodes, $|H|_i = l$, and the maximum size of $\mathcal{Y}_h$ is $d_i = c^2$ if each substructure of a pair can be assigned one of $c$ class. $c$ may be extremely large in some practical applications, for instance, in part-of-speech tagging. We further consider the Hamming loss $L(y, y') = \frac{1}{l}\sum_{k=1}^l \mathbb{I}_{y_k \neq y_k'}$ as in [60, 18]. If we apply Corollary 1 to the pairwise Markov network and divide the bound through by $l$ to normalize the loss as in [60, 18], we obtain generalization bounds of $\mathcal{O}\left(\frac{\ln n}{\rho\sqrt{n}}\left(\log^{\frac{1}{2}}(n^2) + \log^{\frac{1}{2}}(lc)\right) + \sqrt{\frac{\log 1/\delta}{n}}\right)$, which has a logarithmic order dependency on the output space size.

(2.) Consider the special case of structured prediction: multi-class classification, we have $|H|_i = 1$ and $d_i = c$, where $c$ is the number of classes. $c$ may be extremely large since many challenging applications, such as photo and video annotation and web page categorization, can involve tens or hundreds of thousands of classes [62]. For instance, practical web page categorization datasets *AmazonTitles-3M* and *Amazon-3M* have 2,812,281 labels, which are much larger than the training samples (please refer to [5]).

## 3.2 Smoothness Dependent Case

**Assumption 2** *Assume that $\ell$ is $\beta$-smooth, that is*
$$|\nabla\ell(s) - \nabla\ell(t)| \leq \beta|s - t|.$$

Assumption 2 is a pretty mild condition. Both the square hinge loss $\ell(\rho_f) = (1 - \rho_f)_+^2$ and the square margin loss $\ell(\rho_f) = (1_{\rho_f \leq 0} + (1 - \rho_f\rho^{-1})1_{0 < \rho_f \leq \rho})^2$ satisfy Assumption 2 [36].

**Theorem 2** *Under Assumption 2, $\forall v > \max(1, \frac{\sqrt{2}}{2M})$, for any $\delta > 0$, with probability $1 - \delta$ over the sample S of size $n$, the following holds for all $f \in \mathcal{F}$,*

$$R(f) \leq R(\ell(\rho_f)) \leq \max \left\{ \frac{v}{v-1} \hat{R}(\ell(\rho_f)), \hat{R}(\ell(\rho_f)) \right.$$
$$\left. + \mathcal{O}\left( \frac{\beta s^2 \log^4 n}{n} \left( \log(n^2) + \log(sk) \right) + \frac{\log(\frac{1}{\delta})}{n} \right) \right\},$$

*where $s = \max_{i \in [n]} |H_i|$, $k = \max_{i \in [n]} |H_i| d_i$, and where $d_i = \max_{h \in |H_i|} |\mathcal{Y}_h|$.*

**Remark 4** [**Sketch of Theorem 2.**] Theorem 2 suggests that the generalization bound of structured prediction is of order $\mathcal{O}(\frac{s^2}{n})$ when the surrogate loss is smooth w.r.t. the margin function (we hide the logarithmic term). Since the sample size $n$ is typically far larger than $s$, this bound is clearly sharper than bounds in Theorem 1. Compared with [18] whose bounds are of order $\frac{\sqrt{\max_{i \in [n]} |H_i|^2 d_i}}{\sqrt{n}}$ where $d_i = \max_{h \in |H_i|} |\mathcal{Y}_h|$, our generalization bound in Theorem 2 is of order $\frac{\max_{i \in [n]} |H_i|^2}{n}$. And if we consider the case $d_i \geq \max_{i \in [n]} |H_i|^2$ which is very possible since the label set cardinality $d_i$ can be far larger than the factor size $|H_i|$, our bound is sharper than bounds in [18] whether in terms of the sample size $n$ or the order of the numerator. Compared with [51], our bound has a linear convergence rate on the sample size $n$, which is also faster than the slower order $\mathcal{O}(\frac{1}{\sqrt{n}})$ in [51]. Overall, Theorem 2 obtains a sharper bound not only with a faster convergence rate on the sample size $n$ but with the logarithmic dependency on the label set cardinality. We now compare our results with the stability-based results. [41, 42, 40] use the stability tool to investigate the generalization bounds of structure prediction. Denote the number of examples as $n$ and the size of each instance as $m$, under suitable assumptions and hidden logarithmic terms, the latest paper [40] provides the generalization bounds of $\mathcal{O}(\frac{1}{\sqrt{nm}})$ by the stability and PAC-Bayes combination. Compared with this bound, our generalization bounds decrease faster on the sample size $n$. If we consider the case of $n > m$, our bounds are tighter and have a faster convergence rate. In addition, our results are based on the factor graph decomposition approach, which is beneficial for providing the explicit dependency on the properties of the factor graph and helps us show the dependence on the number of possible labels explicitly. And our theoretical analysis reveals that structured prediction can be generalized well even if it has a vast output space and shows that there can be a tighter generalization bound under what conditions. We further compare our bounds with PAC-Bayesian-based bounds. In a seminal work, [46] provides PAC-Bayesian margin guarantees for the simplest classification task, binary classification. The generalization bounds in Corollary 1 and Theorem 3 in [46] are of slow order $\widetilde{\mathcal{O}}(\frac{1}{\sqrt{n}})$. The authors then use the PAC-Bayesian theorem to provide generalization bounds for structured prediction tasks involving language models [45]. [26, 4, 22] further extend [45] to more complex learning scenarios of structured prediction: [26] extends [45] to the maximum loss over random structured outputs; [4] extends [45] to Maximum-A-Posteriori perturbation models; [22] extends [45] by including latent variables. However, regarding the sample size $n$, generalization bounds in the related work [45, 26, 4, 22] are all stated in slow order $\widetilde{\mathcal{O}}(\frac{1}{\sqrt{n}})$. Their analysis typically build on the looser form of McAllester's PAC-Bayesian bound [46] and the global Rademacher complexity [2], which leads to the slow order generalization bound. To our best knowledge, how to use the PAC-Bayesian framework to establish fast rates for structured prediction is still unexplored. It would be very interesting to derive sharper PAC-Bayesian guarantees for structured prediction since a salient advantage of PAC-Bayes is that this theory requires little assumptions [24]. The complete proof of Theorem 2 is provided in Appendix D. Besides, the smooth constant $\beta$ is also typically small. We provide Application 3 for examples in the following.

**Application 3.** Consider the square hinge loss

$$\ell_{sh}(\rho_f(x,y)) = \left( 1 - \left( f(x,y) - \max_{y' \neq y} f(x,y') \right) \right)_+^2,$$

which is $\beta = 2$ smooth w.r.t. the margin $\rho_f$, thus the generalization bound of $\ell_{sh}$ for structured prediction is immediate.

**Remark 5** We now compare our results with the implicit embedding-based results [12, 13, 53, 52, 8, 11, 58, 7]. They mostly provide the convergence rate of slow order $\mathcal{O}(\frac{1}{n^{1/4}})$ w.r.t. the sample

size $n$. The sharpest bound among the related work is provided by [8], which considers the implicit embedding framework from [12], and leverages the fact that learning a discrete output space is easier than learning a continuous counterpart, deriving refined calibration inequalities. Then, [8] uses exponential concentration inequalities to turn the refined results into fast rates under the generalized Massart's or Tsybakov margin condition. Firstly, regarding the sample size $n$, Theorem 6 in [8] shows a excess risk bound of $\mathcal{O}(n^{-\frac{1+\alpha}{2}})$, where $\alpha > 0$ and is a parameter of generalized Tsybakov margin condition, characterizing the hardness of the discrete problem. Although [8] proposes this interesting generalization bound, the bound is presented in expectation. By comparison, our result is presented in high probability, which is beneficial to understand the generalization performance of the learned model when restricted to samples as compared to the rates in expectation. Secondly, generalization error bounds studied in [8] require stronger assumptions. For instance, Theorem 6 is proposed under Bilinear loss decomposition condition, exponential concentration inequality condition, generalized Tsybakov margin condition, bounded loss, and finite prediction space condition. Compared with [8], our bounds are presented without the margin conditions. Recently, there has been some work devoted to providing fast classification rates without standard margin conditions in binary classification, for instance, [6]. Moreover, our Lipschitz continuity condition and smoothness condition are assumed on the margin function. It is easy to construct a surrogate loss which is Lipschitz continuous or smooth w.r.t. the margin function. Finally, our generalization bounds are built upon the factor graph decomposition approach, which is beneficial for providing the explicit dependency on the number of possible labels. In our analysis, we not only improve the dependence on the sample size $n$, but also the dependence on the output space size, explaining that structured prediction can still generalize well in the extremely large output space.

### 3.3 Capacity Dependent Case

**Definition 1 (Covering Number [70, 29])** *Let $\mathcal{F}$ be class of real-valued fucntions, defined over a space $\mathcal{Z}$ and $S := \{\mathbf{z}_1, ..., \mathbf{z}_n\} \in \mathcal{Z}^n$ of cardinality $n$. For any $\epsilon > 0$, the empirical $\ell_\infty$-norm covering number $\mathcal{N}_\infty(\epsilon, \mathcal{F}, S)$ w.r.t $S$ is defined as the minimal number $m$ of a collection of vectors $\mathbf{v}^1, ..., \mathbf{v}^m \in \mathbb{R}^n$ such that ($\mathbf{v}_i^j$ is the $i$-th component of the vector $\mathbf{v}^j$)*

$$\sup_{f \in \mathcal{F}} \min_{j=1,...,m} \max_{i=1,...,n} |f(\mathbf{z}_i) - \mathbf{v}_i^j| \leq \epsilon.$$

*In this case, we call $\{\mathbf{v}^1, ..., \mathbf{v}^m\}$ an $(\epsilon, \ell_\infty)$-cover of $\mathcal{F}$ w.r.t. $S$. Furthermore, the following covering number is introduced:*

$$\mathcal{N}(\epsilon, \mathcal{F}, \|\cdot\|_\infty) := \sup_n \sup_S \mathcal{N}_\infty(\epsilon, \mathcal{F}, S).$$

Before presenting Theorem 3, we use the property $\Psi(x, y) = \sum_{h \in H} \Psi_h(x, y_h)$ to decompose $\Psi(x, y)$ and introduce the following space (different from $\mathcal{F}$ defined in (1)):

$$\mathcal{F}_h := \{\Psi_h(x, y_h) \mapsto \langle w, \Psi_h(x, y_h) \rangle : w \in \mathbb{R}^N, \|w\|_p \leq \Lambda_p\}. \tag{2}$$

Assumptions 3 and 4 are assumptions w.r.t. the covering number bounds (space capacity) on $\mathcal{F}_h$.

**Assumption 3 (polynomial covering number)** *We assume that the function class defined in (2) satisfies that*

$$\log \mathcal{N}(\epsilon, \mathcal{F}_h, \|\cdot\|_\infty) \leq \frac{\gamma_p}{\epsilon^p} \quad \epsilon > 0,$$

*where $0 < p < 2$ and $\gamma_p$ is a positive constant dependent on $p$.*

**Assumption 4 (logarithmic covering number)** *We assume that the function class defined in (2) satisfies that*

$$\log \mathcal{N}(\epsilon, \mathcal{F}_h, \|\cdot\|_\infty) \leq D \log^p(\frac{\gamma}{\epsilon}) \quad \epsilon > 0,$$

*where $p$, $D$ and $\gamma$ are three positive numbers.*

We take the linear function class and the kernel function class for instances to demonstrate the two assumptions. Theorem 4 in [69] and Corollary 9 in [28] give the covering number bound of linear

function class, which satisfies Assumption 3 with $p = 2$. Lemma 19 in [48] extends Theorem 4 in [69] to reproducing kernel Hilbert space, which satisfies Assumption 3 with $p = 2$. If the linear function class is specified by the commonly used Gaussian kernel, the covering number bounds in Theorem 15 of [48] and Theorem 5 of [21] satisfy Assumption 3 with $0 < p \leq 1$. For more covering number bounds of linear function classes, please refer to [64]. Application 4 in the following provides a covering number bound satisfying Assumption 4 if the parameter $w$ is restricted to a euclidean ball.

**Remark 6** Classes that fulfill Assumption 3 are known as satisfying the uniform entropy condition [63]. The popular RKHSs of Gaussian, polynomial and finite rank kernels satisfy the polynomial covering number assumption [23]. Many popular function classes satisfy the logarithmic covering number assumption when the hypothesis class is bounded: Any function space with finite VC-dimension [63, 54], including linear functions and univariate polynomials of degree $k$ (for which $d = k + 1$, and $p = 1$) as special cases; Any RKHS based on a kernel with rank $D$ when $p = 1$ [9]; Any unit ball $\mathcal{B} \subset \mathbb{R}^D$ with fixed $\epsilon \in (0, 1)$ [57].

**Theorem 3** *With different space capacity assumptions, we have the following different results:*

*1) Under Assumptions 1 and 3, $\forall v > \max(1, \frac{\sqrt{2}}{2M})$, for any $\delta > 0$, with probability $1 - \delta$ over the sample $S$, we have*

$$R(f) \leq R(\ell(\rho_f)) \leq \max \left\{ \frac{v}{v-1} \hat{R}(\ell(\rho_f)), \hat{R}(\ell(\rho_f)) + \mathcal{O} \left( \frac{\gamma_p \mu^p s^p}{n^{\frac{2}{p+2}}} + \frac{\log(\frac{1}{\delta})}{n} \right) \right\},$$

*for any $f \in \mathcal{F}$, where $0 < p < 2$, $s = \max_{i \in [n]} |H_i|$, and $\gamma_p$ is a positive constant dependent on $p$.*

*2) Under Assumptions 1 and 4, $\forall v > \max(1, \frac{\sqrt{2}}{2M})$, for any $\delta > 0$, with probability $1 - \delta$ over the sample $S$, we have*

$$R(f) \leq R(\ell(\rho_f)) \leq \max \left\{ \frac{v}{v-1} \hat{R}(\ell(\rho_f)), \hat{R}(\ell(\rho_f)) + \mathcal{O} \left( \frac{D \log^p(\mu s \gamma \sqrt{n})}{n} + \frac{\log(\frac{1}{\delta})}{n} \right) \right\},$$

*for any $f \in \mathcal{F}$, where $p > 0$ and $s = \max_{i \in [n]} |H_i|$, and where $p$, $D$ and $\gamma$ are three positive numbers.*

**Remark 7** [**Sketch of Theorem 3.**] Theorem 3 suggests that, when the function space $\mathcal{F}_h$ satisfies the specific space capacity assumption, the generalization bound of structured prediction can have no dependency on the cardinality of the label set, just presents the dependency on the factor size: $s^p$ or $\log^p(s)$. It also implies that the factor node size $s$ can presents a lower order $\mathcal{O}(s^p)$ (when $0 < p < 1$) or $\mathcal{O}(\log^p(s))$ than linear dependency $\mathcal{O}(s)$ in Theorem 1 or square dependency $\mathcal{O}(s^2)$ in Theorem 2. From Theorem 3, one can also see that, under the polynomial covering number assumption, this bound is sharper than results in Theorem 1 w.r.t. the sample size $n$ when $0 < p < 2$. Under the logarithmic covering number assumption, the bound presents faster order $\mathcal{O}(\frac{1}{n})$ w.r.t. the sample size $n$. Overall, Theorem 3 obtains generalization bounds that have no dependency on the cardinality of the label set and have faster convergence rates than $\mathcal{O}(\frac{1}{\sqrt{n}})$ simultaneously. Theorem 3 in [51] also provides a generalization bound not dependent on the label space size via algorithmic stability. However, their bound requires a convex surrogate loss w.r.t. the parameter $w$ and considering the regularization, so that the regularized empirical risk is strongly convex w.r.t. $w$, which may be restrictive for margin-based surrogate losses [18]. Moreover, this bound is derived in expectation. As a comparison, our bound is obtained with high probability. The proof of Theorem 3 is provided in Appendix E. Besides, as showed in Corollary 2, the Lipschitz continuity constant $\mu$ is typically small. An example satisfying Assumption 4 is given in Application 4.

**Application 4.** We assume that any $\|\Psi_h(x, y_h)\|$ is bounded by $B$, which is a standard assumption in structured prediction. For example, [18] assumes $\max_{i,h,y_h} \|\Psi_h(x, y_h)\|_2 \leq r_2$ where $r_2$ is a constant. Thus, there holds that $|\langle w_1, \Psi_h(x, y_h) \rangle - \langle w_2, \Psi_h(x, y_h) \rangle| \leq B \|w_1 - w_2\|$ for any $w_1, w_2$ and sample $(x, y) \in \mathcal{X} \times \mathcal{Y}$, which implies that $\langle w, \Psi_h(x, y_h) \rangle$ is $B$-Lipschitz w.r.t. $w$. Assume that for any $w \in \mathcal{W}$, $\mathcal{W} \subset \mathbb{R}^N$ is an unit ball. According to Lemma 1.1.8 in [57], any unit ball has the logarithmic covering number bound. Therefore, combined with the Lipschitz assumption, we have

$$\log \mathcal{N}(\epsilon, \mathcal{F}_h, \|\cdot\|_\infty) \leq N \log \left( \frac{3B}{\epsilon} \right), \tag{3}$$

which satisfies Assumption 4. Substituting (3) into Theorem 3, the generalization bound of structured prediction in this setting is immediate.

**Remark 8** [**Sketch of Application 4.**] To compare our bounds with the extensively studied multi-class classification task where $|H_i| = 1$, $d_i = c$, and where $c$ is the number of classes of multi-class classification task, we consider our bounds in the multi-class classification case, which is $\mathcal{O}(\frac{N\log(\mu\sqrt{n})}{n})$. To the best of our knowledge, this is the first high probability bound not dependent on the label size $c$ in multi-class classification. In related work of multi-class classification [34, 33, 36, 55, 50, 65, 48, 49, 44, 18, 35], [18] shows a convergence rate of order $\mathcal{O}(\frac{c}{\sqrt{n}})$ under 2-norm regularization condition; [34] shows a convergence rate of order $\mathcal{O}(\frac{\log^2(nc)}{\sqrt{n}})$ under $\ell_\infty$-norm Lipschitz continuous condition; [33] shows a convergence rate of order $\mathcal{O}(\sqrt{\frac{c}{n}})$ under $\ell_2$-norm Lipschitz continuous condition; [36] shows a convergence rate of order $\mathcal{O}(\frac{\sqrt{c}\log^3(n)}{n})$ under $\ell_2$-norm Lipschitz continuous, smoothness and low noise conditions; [35] shows a convergence rate of order $\mathcal{O}(\frac{\sqrt{c}}{n})$ under $\ell_2$-norm Lipschitz continuous and decay of singular value conditions; [65] shows a convergence rate of order $\mathcal{O}(\frac{\log^3(nc)}{n})$ under $\ell_\infty$-norm Lipschitz continuous and strong convexity conditions (Due to length limit, we just compare our result with the recent work). However, our result shows that if the loss is $\mu$-Lipschitz continuous w.r.t the margin function and the conditions in Application 4 are satisfied, the generalization bound of multi-class classification can show a fast convergence rate of order $\mathcal{O}(\frac{N\log(\sqrt{n})}{n})$. Note that in this result, we do not assume strong curvature conditions: strong convexity or smoothness. If we consider the case $N\log(\sqrt{n}) \leq \log^3(nc)$, our bound is also sharper than the state-of-the-art generalization bound in multi-class classification, under much milder assumptions.

**Excess Risk Bounds.** Define $\hat{f}^* = \arg\min_{f\in\mathcal{F}} \hat{R}(\ell(\rho_f))$ and $f^* = \arg\min_{f\in\mathcal{F}} R(\ell(\rho_f))$, the excess risk of structured prediction is $R(\ell(\rho_{\hat{f}^*})) - R(\ell(\rho_{f^*}))$, which demonstrates the performance of the empirical risk minimizer learned on the samples on the population level and is also an important measure for understanding the generalization performance [2, 38, 39, 66, 37, 67]. Assume that $R(\ell(\rho_f) - \ell(\rho_{f^*}))^2 \leq BR(\ell(\rho_f) - \ell(\rho_{f^*}))$ for some $B > 0$ and every $f \in \mathcal{F}$, we have the following Corollary.

**Corollary 3** *With different space capacity assumptions, we have the following different results:*

*1) Under Assumptions 1 and 3, for any $\delta > 0$, with probability $1 - \delta$ over the sample $S$ of size $n$, we have*

$$R(f) \leq R(\ell(\rho_{\hat{f}^*})) \leq R(\ell(\rho_{f^*})) + \mathcal{O}\left(\frac{\gamma_p \mu^p s^p}{n^{\frac{2}{p+2}}} + \frac{\log(\frac{1}{\delta})}{n}\right).$$

*for any $f \in \mathcal{F}$, where $0 < p < 2$, $s = \max_{i\in[n]} |H_i|$, and $\gamma_p$ is a positive constant dependent on $p$.*

*2) Under Assumptions 1 and 4, for any $\delta > 0$, with probability $1 - \delta$ over the sample $S$ of size $n$, we have*

$$R(f) \leq R(\ell(\rho_{\hat{f}^*})) \leq R(\ell(\rho_{f^*})) + \mathcal{O}\left(\frac{\log^p(\mu s\gamma\sqrt{n})}{n} + \frac{\log(\frac{1}{\delta})}{n}\right).$$

*for any $f \in \mathcal{F}$, where $p > 0$, $s = \max_{i\in[n]} |H_i|$, and $p$, $D$ and $\gamma$ are three positive numbers.*

**Remark 9** [**Sketch of Corollary 3.**] The excess risk bounds in Corollary 3 have the same order convergence rates as results in Theorem 3. And the meaning of these results has been discussed in Remark 7 and Remark 8. We provide the complete proof of Corollary 3 in Appendix F. Considering the length limit, some discussion of this paper is postponed to the appendix.

## 4 Conclusion

In this paper, we are towards sharper generalization bounds for structured prediction. The analysis is based on the factor graph decomposition approach, which can help shed new light on the role played by the graph in generalization. We present state-of-the-art generalization bounds from different perspectives. Overall, the bounds presented have answered the three questions posed in Section 1. We believe our theoretical findings can provide deep insights into the learning guarantees of structured prediction. Additionally, we are also concerned about whether the convergence rate of structured prediction can reach faster order than $\mathcal{O}(1/n)$? We will investigate this problem in future work and design new algorithms based on our theoretical analysis.

## Acknowledgment

We appreciate all the anonymous reviewers, ACs, and PCs for their invaluable and constructive comments. This work is supported in part by the National Natural Science Foundation of China (No. 62076234, No.61703396, No. 62106257), Beijing Outstanding Young Scientist Program NO.BJJWZYJH012019100020098, Intelligent Social Governance Platform, Major Innovation & Planning Interdisciplinary Platform for the "Double-First Class" initiative, Renmin University of China, China Unicom Innovation Ecological Cooperation Plan, Public Computing Cloud of Renmin University of China.

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
