# Supplementary Material for "Towards Sharper Generalization Bounds for Structured Prediction"

**Shaojie Li[1,2]**    **Yong Liu[1,2,*]**
[1]Gaoling School of Artificial Intelligence, Renmin University of China, Beijing, China
[2]Beijing Key Laboratory of Big Data Management and Analysis Methods, Beijing, China
2020000277@ruc.edu.cn, liuyonggsai@ruc.edu.cn

## Abstract

In this supplementary material, we provide complete proofs of the theorems of the main paper.

## A    Proof of Theorem 1

### A.1    Preliminaries

In Section 2 of the main paper, the loss function space is defined as:

$$\mathcal{L}_\rho = \{\ell_\rho(x, y, f) := \ell(\rho_f(x, y)) : f \in \mathcal{F}\}, \tag{1}$$

where

$$\mathcal{F} = \{x \mapsto \langle w, \Psi(x, y) \rangle : w \in \mathbb{R}^N, \|w\|_p \le \Lambda_p\}.$$

We now introduce the function space of the margin function

$$\widetilde{\mathcal{F}}_\rho = \{(x, y) \mapsto \rho_f(x, y) : f \in \mathcal{F}\}. \tag{2}$$

We also denote $\rho_f(x, y)$ as $\rho(x, y, f)$ and introduce the Rademacher complexity definition of the loss function space:

**Definition 1** (Rademacher Complexity [2]). *Assume $\mathcal{L}_\rho$ is a space of loss functions as defined in equation (1), then the empirical Rademacher complexity of $\mathcal{L}_\rho$ is:*

$$\hat{\Re}(\mathcal{L}_\rho) = \mathbb{E}_{\boldsymbol{\sigma}} \left[ \sup_{\ell_\rho \in \mathcal{L}_\rho} \frac{1}{n} \sum_{i=1}^n \sigma_i \ell_\rho(x_i, y_i, f) \right],$$

*where $\sigma_1, \sigma_2, ..., \sigma_n$ are i.i.d. Rademacher variables taking values $-1$ and $1$ with equal probability, independent of the sample $S = \{(x_1, y_1), ..., (x_n, y_n)\}$. The Rademacher complexity of $\mathcal{L}_\rho$ is $\Re(\mathcal{L}_\rho) = \mathbb{E}_{(x,y) \sim P} \hat{\Re}(\mathcal{L}_\rho)$, where $P$ is the underlying distribution.*

Besides, we define the empirical risk of any scoring function $f$ as

$$\hat{R}(\ell_\rho) = \frac{1}{n} \sum_{i=1}^n \ell_\rho(x_i, y_i, f),$$

and the expected risk is defined as

$$R(\ell_\rho) = \mathbb{E}_{(x,y) \sim P}[\ell_\rho(x, y, f)].$$

---

*Corresponding Author.

35th Conference on Neural Information Processing Systems (NeurIPS 2021).

According to the McDiarmid inequality [8] and the symmetrization technique (e.g., Theorem 4.4 in [10]), it is easy to obtain that with probability at least $1 - \delta$,

$$R(\ell_\rho) - \hat{R}(\ell_\rho) \leq 2\hat{\Re}(\mathcal{L}_\rho) + 3M\sqrt{\frac{\log 1/\delta}{2n}}.$$

Due to Lemma A.4 in [5], combined with the Lipschitz property of $\ell_\rho$ in the Assumption 1 of the main paper, we have the following inequality with probability at least $1 - \delta$:

$$R(\ell_\rho) - \hat{R}(\ell_\rho) \leq 2\mu\hat{\Re}(\widetilde{\mathcal{F}}_\rho) + 3M\sqrt{\frac{\log 1/\delta}{2n}}. \tag{3}$$

Thus the key step is to bound the term $\hat{\Re}(\widetilde{\mathcal{F}}_\rho)$.

## A.2   Covering number bound

To bound the term $\hat{\Re}(\widetilde{\mathcal{F}}_\rho)$, we first introduce the definition of covering number.

**Definition 2** (Covering Number [15]). *Let $\mathcal{F}$ be class of real-valued fucntions, defined over a space $\mathcal{Z}$ and $S := \{\mathbf{z}_1, ..., \mathbf{z}_n\} \in \mathcal{Z}^n$ of cardinality $n$. For any $\epsilon > 0$, the empirical $\ell_\infty$-norm covering number $\mathcal{N}_\infty(\epsilon, \mathcal{F}, S)$ w.r.t $S$ is defined as the minimal number $m$ of a collection of vectors $\mathbf{v}^1, ..., \mathbf{v}^m \in \mathbb{R}^n$ such that ($\mathbf{v}_i^j$ is the $i$-th component of the vector $\mathbf{v}^j$)*

$$\sup_{f \in \mathcal{F}} \min_{j=1,...,m} \max_{i=1,...,n} |f(\mathbf{z}_i) - \mathbf{v}_i^j| \leq \epsilon.$$

*In this case, we call $\{\mathbf{v}^1, ..., \mathbf{v}^m\}$ an $(\epsilon, \ell_\infty)$-cover of $\mathcal{F}$ w.r.t. $S$. We denote $\mathcal{N}_\infty(\epsilon, \mathcal{F}, n) = \sup_S \mathcal{N}_\infty(\epsilon, \mathcal{F}, S)$. Furthermore, the following covering number is introduced:*

$$\mathcal{N}(\epsilon, \mathcal{F}, \|\cdot\|_\infty) := \sup_n \sup_S \mathcal{N}_\infty(\epsilon, \mathcal{F}, S).$$

We then need to introduce the following lemmas.

**Lemma 1** ([14]). *Let $\mathcal{L}$ be a class of linear functions. If $\|\mathbf{x}\|_p \leq b$ and $\|\mathbf{w}\|_q \leq a$, where $2 \leq p < \infty$ and $1/p + 1/q = 1$, then $\forall \epsilon > 0$,*

$$\log \mathcal{N}_\infty(\epsilon, \mathcal{L}, n) \leq 36(p-1)\frac{a^2 b^2}{\epsilon^2} \log_2\left[2\lceil 4ab/\epsilon + 2\rceil n + 1\right],$$

*where $\mathcal{N}_\infty(\epsilon, \mathcal{L}, n) = \sup_S \mathcal{N}_\infty(\epsilon, \mathcal{L}, S)$, and where $S = \{(x_1, y_1), ..., (x_n, y_n)\} \in (\mathcal{X} \times \mathcal{Y})^n$.*

**Lemma 2.** *For any scoring function $f$ and $\widetilde{f}$, and any sample $(x_i, y_i)$, we have the following property:*

$$\left|\rho_f(x_i, y_i, f) - \rho_{\tilde{f}}(x_i, y_i, \tilde{f})\right| \leq 2 \sum_{h \in H_i} \left|\max_{y \in \mathcal{Y}_h} f_h(x_i, y) - \widetilde{f}_h(x_i, y)\right|,$$

*Proof.* Based on the notations, we have

$$\left|\rho_f(x_i, y_i, f) - \rho_{\tilde{f}}(x_i, y_i, \tilde{f})\right|$$

$$\leq \left|f(x_i, y_i) - \max_{y' \neq y_i} f(x_i, y') - \tilde{f}(x_i, y_i) + \max_{y' \neq y_i} \tilde{f}(x_i, y')\right|$$

$$\leq \left|\left[\max_{y' \neq y_i} f(x_i, y_i) - \widetilde{f}(x_i, y_i)\right]\right| + \left|\left[\max_{y' \neq y_i} f(x_i, y') - \widetilde{f}(x_i, y')\right]\right|$$

$$\leq 2\left|\max_{y \in \mathcal{Y}} f(x_i, y) - \widetilde{f}(x_i, y)\right|$$

$$\leq 2 \sum_{h \in H_i} \left|\max_{y \in \mathcal{Y}} f_h(x_i, y_h) - \widetilde{f}_h(x_i, y_h)\right|$$

$$= 2 \sum_{h \in H_i} \left|\max_{y \in \mathcal{Y}_h} f_h(x_i, y) - \widetilde{f}_h(x_i, y)\right|.$$

The proof is over. □

The following proposition is the covering number bound on the margin function class $\widetilde{\mathcal{F}}_\rho$.

**Proposition 1.** *For the function class $\widetilde{\mathcal{F}}_\rho$ defined in (2), we have*

$$\log \mathcal{N}_\infty(\epsilon, \widetilde{\mathcal{F}}_\rho, S) \leq \frac{144(q-1)s^2 \Lambda_p^2 r_q^2}{\epsilon^2} \log \left[ 2 \left\lceil 8 \frac{s\Lambda_p r_q}{\epsilon} + 2 \right\rceil k + 1 \right],$$

*where $s = \max_{i \in [n]} |H_i|$, $2 \leq p < \infty$, $1/p + 1/q = 1$, $r_q = \max_{i,h,y} \|\Psi_h(x_i, y)\|_q$, and $k = \sum_{i \in [n]} \sum_{h \in |H|_i} \sum_{y \in \mathcal{Y}_h}$.*

***Proof of Proposition 1.*** The proof is inspired by [6]. A difficulty towards this aim consists in the non-linearity of margin $\rho_f$. We bypass this obstacle by introducing the following linear function class:

$$\widetilde{F} := \{ v \mapsto \langle w, v \rangle : w \in \mathbb{R}^N, \|w\|_p \leq \Lambda_p, v \in \widetilde{S} \}, \tag{4}$$

where $\widetilde{S}$ is defined as follows

$$\widetilde{S} := \{ \Psi_h(x, y) : x \in \{x_1, ..., x_n\}, h \in H_i, y \in \mathcal{Y}_h \}. \tag{5}$$

We relate the covering number of the non-linearity function class $\widetilde{\mathcal{F}}_\rho$ to the covering number of this linear function $\widetilde{F}$. The latter is easy to be addressed since it is a linear function class, to which standard arguments apply, such as Lemma 1.

We now relate the empirical $\ell_\infty$-norm convering numbers of $\widetilde{F}$ w.r.t. $\widetilde{S}$ to that of $\widetilde{\mathcal{F}}_\rho$ w.r.t. $S$. Let

$$\left\{ \mathbf{r}^j = (r^j_{1,1,1}, ..., r^j_{1,h_{|H_1|},|\mathcal{Y}_{h_{|H_1|}}|}, r^j_{2,1,1}, ..., r^j_{2,h_{|H_2|},|\mathcal{Y}_{h_{|H_2|}}|}, ..., r^j_{n,1,1}, ..., r^j_{n,h_{|H_n|},|\mathcal{Y}_{h_{|H_n|}}|}) : j = 1, ..., N \right\}$$

be an $(\epsilon, \ell_\infty)$-cover of $\widetilde{F}$ with $N$ be the cardinality. That is, for any $w \in \mathbb{R}^N$ with $\|w\|_p \leq \Lambda_p$, this cover guarantees the existence of $j_{(w)} \in \{1, ..., N\}$ such that

$$\max_{i \in [n]} \max_{h \in H_i} \max_{y \in \mathcal{Y}_h} \left| r^{j_{(w)}}_{i,h,y} - \langle w, \Psi_h(x_i, y) \rangle \right| \leq \epsilon. \tag{6}$$

Now we define $\mathbf{r}^j_i = (\sum_{h \in H_i} r^j_{i,h,\tau})$ for all $j \in [N]$, where $\tau$ represents labels in the label space of factor $h$ and is corresponding to $y_h$ in the term $\Psi_h(x, y_h)$. Take sample $i = 1$ for example, $(\sum_{h \in H_1} r^j_{1,h,\tau})$ is a vector where $\tau$ choose elements from the label space of factor $h$. It is important to note that if $\mathbf{r}^j_i$ is assigned with sample $y_i$ (such as the following $\rho_{\mathbf{r}^j_i}(x_i, y_i, \mathbf{r}^j_i)$), then it becomes an element whose $\tau$ corresponds to sample $y_i$, so that we can use the Lipschitz property established in Lemma 2 to bound the difference of the margin function by the scoring function on the factor level. Therefore, we define the set

$$\{ (\rho_{\mathbf{r}^j_1}(x_1, y_1, \mathbf{r}^j_1), \rho_{\mathbf{r}^j_2}(x_2, y_2, \mathbf{r}^j_2), ..., \rho_{\mathbf{r}^j_n}(x_n, y_n, \mathbf{r}^j_n)) : j = 1, ..., N \}. \tag{7}$$

Then we have:

$$\max_{i \in [n]} \left| \rho_f(x_i, y_i, f) - \rho_{\mathbf{r}^{j_{(w)}}_i}(x_i, y_i, \mathbf{r}^{j_{(w)}}_i) \right|$$

$$\leq \max_{i \in [n]} 2 \sum_{h \in H_i} \left| \max_{y \in \mathcal{Y}_h} f_h(x_i, y) - r^{j_{(w)}}_{i,h,y} \right| \quad \text{Using Lemma 2}$$

$$\leq 2 \max_{i \in [n]} \max_{h \in H_i} \max_{y \in \mathcal{Y}_h} |H_i| \left| \langle w, \Psi_h(x_i, y) \rangle - r^{j_{(w)}}_{i,h,y} \right|$$

$$\leq 2 \max_{i \in [n]} |H_i| \epsilon \quad \text{Using (6).}$$

Denote $s$ by $\max_{i \in [n]} |H_i|$. The above analysis shows that the set defined in (7) is also an $(2s\epsilon, \ell_\infty)$-cover of $\widetilde{\mathcal{F}}_\rho$ w.r.t $S = \{(x_1, y_1), ..., (x_n, y_n)\}$. Therefore,

$$\log \mathcal{N}_\infty(\epsilon, \widetilde{\mathcal{F}}_\rho, S) \leq \log \mathcal{N}_\infty(\frac{1}{2s}\epsilon, \widetilde{F}, \widetilde{S}). \tag{8}$$

Based on Lemma 1, we have

$$\log \mathcal{N}_\infty(\epsilon, \widetilde{F}, \widetilde{S}) \leq \frac{36(q-1)\Lambda_p^2 r_q^2}{\epsilon^2} \log \left[ 2 \left[ 4 \frac{\Lambda_p r_q}{\epsilon} + 2 \right] k + 1 \right], \tag{9}$$

where $s = \max_{i\in[n]} |H_i|$, $2 \leq p < \infty$, $1/p + 1/q = 1$, $r_q = \max_{i,h,y} \|\Psi_h(x_i, y)\|_q$, and $k = \sum_{i\in[n]} \sum_{h\in|H|_i} \sum_{y\in\mathcal{Y}_h}$.

Combined (8) with (9), the proof of Proposition 1 is completed. □

## A.3 Proof of Theorem 1

**Lemma 3** ([4]). *Let $\mathcal{F}$ be a real-valued function class taking values in $[0,1]$, and assume that $0 \in \mathcal{F}$. Let $S$ be a finite sample of size $n$. For any $2 \leq p \leq \infty$, we have the following relationship between the Rademacher complexity $\hat{\Re}(\mathcal{F})$ and the covering number $\mathcal{N}_p(\mathcal{F}, \epsilon, S)$.*

$$\hat{\Re}(\mathcal{F}) \leq \inf_{\alpha>0} \left( 4\alpha + \frac{12}{\sqrt{n}} \int_\alpha^1 \sqrt{\log \mathcal{N}_p(\mathcal{F}, \epsilon, S)} d\epsilon \right). \tag{10}$$

*Proof of Theorem 1.* Denoted by $a := 144(q-1)s^2\Lambda_p^2 r_q^2$, $b := 16s\Lambda_p r_q k$ and $c = 6k + 1$. Based on Lemma 3 and Proposition 1, we have:

$$\hat{\Re}(\widetilde{\mathcal{F}}_\rho) \leq \inf_{\alpha>0} \left( 4\alpha + \frac{12}{\sqrt{n}} \int_\alpha^1 \sqrt{\log \mathcal{N}_\infty(\widetilde{\mathcal{F}}_\rho, \epsilon, S)} d\epsilon \right)$$

$$\leq \inf_{\alpha>0} \left( 4\alpha + \frac{12}{\sqrt{n}} \int_\alpha^1 \frac{\sqrt{a\log(b/\epsilon + c)}}{\epsilon} d\epsilon \right)$$

$$\leq \frac{4}{n} + \frac{12}{\sqrt{n}} \int_{1/n}^1 \frac{\sqrt{a\log(bn + c)}}{\epsilon} d\epsilon$$

$$= \frac{4}{n} + \frac{12\ln n}{\sqrt{n}} \sqrt{a\log(bn + c)}.$$

Substituting this result into (3), with probability at least $1 - \delta$, we have

$$R(\ell_\rho) - \hat{R}(\ell_\rho) \leq \frac{8\mu}{n} + \frac{24\mu \ln n}{\sqrt{n}} \sqrt{144(q-1)s^2\Lambda_p^2 r_q^2 \log(16s\Lambda_p r_q kn + 6k + 1)} + 3M\sqrt{\frac{\log 1/\delta}{2n}},$$

where $s = \max_{i\in[n]} |H_i|$, $2 \leq p < \infty$, $1/p + 1/q = 1$, $r_q = \max_{i,h,y} \|\Psi_h(x_i, y)\|_q$, and $k = \sum_{i\in[n]} \sum_{h\in|H|_i} \sum_{y\in\mathcal{Y}_h}$. That is we have

$$R(f) \leq R(\ell(\rho_f)) \leq \hat{R}(\ell(\rho_f)) + \mathcal{O}\left( \frac{\mu s \ln n}{\sqrt{n}} \log^{\frac{1}{2}}(nsk) + \sqrt{\frac{\log 1/\delta}{n}} \right),$$

By some simple transformations of notations, the conclusion of Theorem 1 in the main paper can be easily verified. The proof is over. □

**Remark 1. [Sketch of proof techniques.]** Our proof is based on space complexity tools: Rademacher complexity [2] and Covering number [15]. According to the McDiarmid inequality [8] and the symmetrization technique (e.g., Theorem 4.4 in [10]), combined with the Lipschitz property of $\ell_\rho$, the key step in our proof is to bound the empirical Rademacher complexity of the margin function class: $\hat{\Re}(\widetilde{\mathcal{F}}_\rho)$. We use the refined Dudley entropy integral inequality in [4] with $\ell_\infty$-norm to bound this term $\hat{\Re}(\widetilde{\mathcal{F}}_\rho)$: $\hat{\Re}(\widetilde{\mathcal{F}}_\rho) \leq \inf_{\alpha>0}(4\alpha + \frac{12}{\sqrt{n}} \int_\alpha^1 \sqrt{\log \mathcal{N}_\infty(\widetilde{\mathcal{F}}_\rho, \epsilon, S)} d\epsilon)$. Then the proof switches to bound the $\ell_\infty$-norm covering number of the margin function class: $\log \mathcal{N}_\infty(\epsilon, \widetilde{\mathcal{F}}_\rho, S)$. The $\ell_\infty$-norm covering number takes advantage of the max operator in the margin, which admits us to improve the dependency on the size of the output space. The challenge lies in that the margin function is nonlinear. To deal with the nonlinear margin function class $\widetilde{\mathcal{F}}_\rho$, we construct a simpler linear function class: $\widetilde{F} := \{v \mapsto \langle w, v \rangle : w \in \mathbb{R}^N, \|w\|_p \leq \Lambda_p, v \in \widetilde{S}\}$, where $\widetilde{S}$ is defined as follows: $\widetilde{S} := \{\Psi_h(x, y) : x \in \{x_1, ..., x_n\}, h \in H_i, y \in \mathcal{Y}_h\}$. The covering number of $\widetilde{F}$ can be connected to the nonlinear margin function class: $\log \mathcal{N}_\infty(\epsilon, \widetilde{\mathcal{F}}_\rho, S) \leq \log \mathcal{N}_\infty(\frac{1}{2s}\epsilon, \widetilde{F}, \widetilde{S})$, while the covering number bound of the linear function $\widetilde{F}$ is easier to handle [14, 6].

## B   Proof of Corollary 1

In [3], they provide two margin loss, additive and the multiplicative empirical margin losses, that can be used to guarantee many existing structured prediction, defined as:

$$\ell_\rho^{add}(x,y,f) := \ell^{add}(\rho_f(x,y)) = \Phi^* \left( \max_{y' \neq y} L(y',y) - \frac{1}{\rho} \left[ f(x,y) - f(x,y') \right] \right),$$

$$\ell_\rho^{mult}(x,y,f) := \ell^{mult}(\rho_f(x,y)) = \Phi^* \left( \max_{y' \neq y} L(y',y) \left( 1 - \frac{1}{\rho} \left[ f(x,y) - f(x,y') \right] \right) \right),$$

where $\Phi^*(r) = \min \left( \max_{y,y'} L(y,y'), \max(0,r) \right)$. To prove Corollary 1, we should show the Lipschitz continuity property of $\ell_\rho^{add}(x,y,f)$ and $\ell_\rho^{mult}(x,y,f)$.

**Lemma 4.** *For any scoring function $f$ and $\widetilde{f}$, and any sample $(x,y)$, we have*

$$\left| \ell_\rho^{add}(x,y,f) - \ell_\rho^{add}(x,y,\widetilde{f}) \right| \leq \frac{1}{\rho} \left| \rho_f(x,y) - \rho_{\widetilde{f}}(x,y) \right|,$$

*and*

$$\left| \ell_\rho^{mult}(x,y,f) - \ell_\rho^{mult}(x,y,\widetilde{f}) \right| \leq \frac{M}{\rho} \left| \rho_f(x,y) - \rho_{\widetilde{f}}(x,y) \right|.$$

*Proof.* Note that $\Phi^*(r)$ is 1-Lipschitz continuous w.r.t. $r$.
(1) For the additive margin loss, we have

$$\left| \ell_\rho^{add}(x,y,f) - \ell_\rho^{add}(x,y,\widetilde{f}) \right|$$

$$\leq \left| \left( \max_{y' \neq y} L(y',y) - \frac{1}{\rho} \left[ f(x,y) - f(x,y') \right] \right) - \left( \max_{y' \neq y} L(y',y) - \frac{1}{\rho} \left[ \widetilde{f}(x,y) - \widetilde{f}(x,y') \right] \right) \right|$$

$$\leq \frac{1}{\rho} \left| \left[ \max_{y' \neq y} \widetilde{f}(x,y) - \widetilde{f}(x,y') \right] - \left[ \max_{y' \neq y} f(x,y) - f(x,y') \right] \right|$$

$$\leq \frac{1}{\rho} \left| \rho_f(x,y) - \rho_{\widetilde{f}}(x,y) \right|.$$

(2) For the multiplicative margin loss, we have

$$\left| \ell_\rho^{mult}(x,y,f) - \ell_\rho^{mult}(x,y,\widetilde{f}) \right|$$

$$\leq \left| \max_{y' \neq y} L(y',y) \left( 1 - \frac{1}{\rho} \left[ f(x,y) - f(x,y') \right] \right) - \max_{y' \neq y} L(y',y) \left( 1 - \frac{1}{\rho} \left[ \widetilde{f}(x,y) - \widetilde{f}(x,y') \right] \right) \right|$$

$$\leq \frac{M}{\rho} \left| \left[ \max_{y' \neq y} \widetilde{f}(x,y) - \widetilde{f}(x,y') \right] - \left[ \max_{y' \neq y} f(x,y) - f(x,y') \right] \right|$$

$$\leq \frac{M}{\rho} \left| \rho_f(x,y) - \rho_{\widetilde{f}}(x,y) \right|.$$

The proof is over. $\qquad \square$

*proof of Corollary 1.* Substituting $\mu = \frac{1}{\rho}$ and $\mu = \frac{M}{\rho}$ for $\ell_\rho^{add}$ and $\ell_\rho^{mult}$ into Theorem 1, respectively, Corollary 1 can be easily verified. $\qquad \square$

## C   Proof of Corollary 2

*Proof.* For a fixed $\mathbf{f} = (f_1,...,f_T)$, any $\alpha$ in the probability simplex $\Delta$ defines a distribution over $\{f_1,...,f_T\}$. Sampling from $\{f_1,...,f_T\}$ according to $\alpha$ and averaging leads to functions $m$ of the form $m = \frac{1}{n'} \sum_{i=1}^T n_t f_t$ for some $\mathbf{n} = (n_1,...,n_T) \in \mathbb{N}^T$, with $\sum_{t=1}^T n_t = n'$, and $f_t \in F_{k_t}$. For any $\mathbf{N} = (N_1,...,N_p)$ with $|\mathbf{N}| = n'$, we consider the family of functions

$$M_{\mathcal{G},\mathbf{N}} = \left\{ \frac{1}{n'} \sum_{k=1}^p \sum_{j=1}^{N_k} f_{k,j} | \forall (k,j) \in [p] \times [N_k], f_{k,j} \in F_k \right\},$$

and the union of all such families $M_{\mathcal{G},n'} = \cup_{|\mathbf{N}|=n'} M_{\mathcal{G},\mathbf{N}}$. Also the margin function class is defined as

$$M_{\rho,\mathcal{G},\mathbf{N}} = \{\rho_m : m \in M_{\mathcal{G},\mathbf{N}}\}.$$

Fix $\rho > 0$. For a fixed $\mathbf{N}$, the empirical Rademacher complexity of $M_{\rho,\mathcal{G},\mathbf{N}}$ can be bounded as follows for any $n' \geq 1$:

$$\hat{\Re}(M_{\rho,\mathcal{G},\mathbf{N}}) \leq \frac{1}{n'} \sum_{k=1}^{p} N_k \hat{\Re}(\widetilde{\mathcal{F}}_{\rho,k}).$$

Thus, by Eq (3), we have the following bound holds: for any $\delta > 0$, with probability at least $1 - \delta$, for all $m \in M_{\mathcal{G},\mathbf{N}}$,

$$R(\ell_{\rho,\tau}(m)) - \hat{R}(\ell_{\rho,\tau}(m)) \leq 2\mu \hat{\Re}(M_{\rho,\mathcal{G},\mathbf{N}}) + 3M \sqrt{\frac{\log 1/\delta}{2n}}$$

$$\leq 2\mu \frac{1}{n'} \sum_{k=1}^{p} N_k \hat{\Re}(\widetilde{\mathcal{F}}_{\rho,k}) + 3M \sqrt{\frac{\log 1/\delta}{2n}}.$$

Since there are at most $p^{n'}$ possible $p$-tuples $\mathbf{N}$ with $|\mathbf{N}| = n'$, by the union bound, for any $\delta > 0$, with probability at least $1 - \delta$, for all $m \in M_{\mathcal{G},n'}$, we can write

$$R(\ell_{\rho,\tau}(m)) - \hat{R}(\ell_{\rho,\tau}(m)) \leq 2\mu \frac{1}{n'} \sum_{k=1}^{p} N_k \hat{\Re}(\widetilde{\mathcal{F}}_{\rho,k}) + 3M \sqrt{\frac{\log p^{n'}/\delta}{2n}}.$$

Thus, with probability at least $1 - \delta$, for all functions $m = \frac{1}{n'} \sum_{i=1}^{T} n_t f_t$ with $f_t \in \mathcal{F}_{k_t}$, the following inequality holds

$$R(\ell_{\rho,\tau}(m)) - \hat{R}(\ell_{\rho,\tau}(m)) \leq 2\mu \frac{1}{n'} \sum_{k=1}^{p} \sum_{t:k_t=k} n_t \hat{\Re}(\widetilde{\mathcal{F}}_{\rho,k_t}) + 3M \sqrt{\frac{\log p^{n'}/\delta}{2n}}.$$

Taking the expectation with respect to $\alpha$ and using $\mathbb{E}_\alpha[n_t/n'] = \alpha_t$, we obtain that for any $\delta > 0$, with probability at least $1 - \delta$, for all $m$, we have

$$\mathbb{E}_\alpha\left[R(\ell_{\rho,\tau}(m)) - \hat{R}(\ell_{\rho,\tau}(m))\right] \leq 2\mu \sum_{t=1}^{T} \alpha_t \hat{\Re}(\widetilde{\mathcal{F}}_{\rho,k_t}) + 3M \sqrt{\frac{\log p^{n'}/\delta}{2n}}.$$

Fix $n' \geq 1$. Then, for any $\delta_{n'} > 0$, with probability at least $1 - \delta_{n'}$,

$$\mathbb{E}_\alpha\left[R(\ell_{\rho,\tau}(m)) - \hat{R}(\ell_{\rho,\tau}(m))\right] \leq 2\mu \sum_{t=1}^{T} \alpha_t \hat{\Re}(\widetilde{\mathcal{F}}_{\rho,k_t}) + 3M \sqrt{\frac{\log p^{n'}/\delta_{n'}}{2n}}.$$

Choose $\delta_{n'} = \frac{\delta}{2p^{n'-1}}$ for some $\delta > 0$, then for $p \geq 2$, $\sum_{n' \geq 1} \delta_{n'} = \frac{\delta}{2(1-1/p)} \leq \delta$. Thus, for any $\delta > 0$ and $n' \geq 1$, with probability at least $1 - \delta$, the following holds for all $m$:

$$\mathbb{E}_\alpha\left[R(\ell_{\rho,\tau}(m)) - \hat{R}(\ell_{\rho,\tau}(m))\right] \leq 2\mu \sum_{t=1}^{T} \alpha_t \hat{\Re}(\widetilde{\mathcal{F}}_{\rho,k_t}) + 3M \sqrt{\frac{\log 2p^{2n'-1}/\delta}{2n}}.$$

We first consider the additive margin loss, defined as

$$\ell_{\rho,\tau}^{add}(x,y,m) := \ell_\tau^{add}(\rho_m(x,y)) = \Phi^*\left(\max_{y' \neq y} L(y',y) + \tau - \frac{1}{\rho}[m(x,y) - m(x,y')]\right).$$

Thus, there holds that

$$\mathbb{E}_\alpha\left[R(\ell_{\rho,1/2}^{add}(m)) - \hat{R}(\ell_{\rho,1/2}^{add}(m))\right] \leq \frac{4}{\rho} \sum_{t=1}^{T} \alpha_t \hat{\Re}(\widetilde{\mathcal{F}}_{\rho,k_t}) + 3M \sqrt{\frac{\log 2p^{2n'-1}/\delta}{2n}}.$$

Now, for any $g = \sum_{t=1}^{T} \alpha_t f_t \in \mathcal{G}$ and any $m = \frac{1}{n'} \sum_{i=1}^{T} n_t f_t$, we have

$$
\begin{aligned}
R(g) &= \mathbb{E}\left[L(\hat{g}(x), y)\right] \\
&= \mathbb{E}\left[L(\hat{g}(x), y)1_{\rho_g(x,y) \leq 0}\right],
\end{aligned}
$$

and the following proof follows the Section A.8 in the Appendix of [3]. For brevity, we omit it here. Following their proof, we can finally obtain that

$$
R(g) - \hat{R}(\ell_{\rho,1}^{add}(g)) \leq \frac{2M}{\rho}\sqrt{\frac{\log p}{n}} + \frac{4}{\rho}\sum_{t=1}^{T} \alpha_t \hat{\Re}(\widetilde{\mathcal{F}}_{\rho,k_t}) + 9M\sqrt{\lceil\frac{4}{\rho^2}\log(\frac{|\mathcal{Y}|^2\rho^2 n}{4\log p})\rceil\frac{\log p}{n} + \frac{\log 2/\delta}{2n}}.
$$

Because for any $k_t \in [1, p]$, in the *proof of Theorem 1* part, there holds that

$$
\hat{\Re}(\widetilde{\mathcal{F}}_{\rho,k_t}) \leq \frac{4}{n} + \frac{12\ln n}{\sqrt{n}}\sqrt{144(q-1)s^2\Lambda_p^2 r_q^2 \log(16s\Lambda_p r_q kn + 6k + 1)}.
$$

Since $\sum_{t=1}^{T} \alpha_t = 1$, we have the following bound:

$$
\begin{aligned}
R(g) - \hat{R}(\ell_{\rho,1}^{add}(g)) \leq\ & 9M\sqrt{\lceil\frac{4}{\rho^2}\log(\frac{|\mathcal{Y}|^2\rho^2 n}{4\log p})\rceil\frac{\log p}{n} + \frac{\log 2/\delta}{2n}} + \frac{2M}{\rho}\sqrt{\frac{\log p}{n}} + \frac{16}{\rho n} \\
& + \frac{4}{\rho}\frac{12\ln n}{\sqrt{n}}\sqrt{144(q-1)s^2\Lambda_p^2 r_q^2 \log(16s\Lambda_p r_q kn + 6k + 1)},
\end{aligned}
$$

where $s = \max_{i\in[n]}|H_i|$, $2 \leq p < \infty$, $1/p + 1/q = 1$, $r_q = \max_{i,h,y}\|\Psi_h(x_i, y)\|_q$, and $k = \sum_{i\in[n]}\sum_{h\in|H|_i}\sum_{y\in\mathcal{Y}_h}$.

That is we have

$$
R(g) - \hat{R}(\ell_{\rho,1}^{add}(g)) \leq \mathcal{O}\left(\frac{s\ln n}{\rho\sqrt{n}}\left(\log^{\frac{1}{2}}(nsk)\right) + \sqrt{C(n, p, \rho, |\mathcal{Y}|, \delta)}\right),
$$

where $C(n, p, \rho, |\mathcal{Y}|, \delta) = \lceil\frac{1}{\rho^2}\log(\frac{|\mathcal{Y}|^2\rho^2 n}{4\log p})\rceil\frac{\log p}{n} + \frac{\log 2/\delta}{n}$.

For the multiplicative margin losses

$$
\ell_{\rho,\tau}^{mult}(x, y, m) := \ell_{\tau}^{mult}(\rho_m(x, y)) = \Phi^*\left(\max_{y'\neq y}L(y', y)\left(1 + \tau - \frac{1}{\rho}\left[m(x, y) - m(x, y')\right]\right)\right),
$$

by a similar proof, we have

$$
\begin{aligned}
R(g) - \hat{R}(\ell_{\rho,1}^{mult}(g)) \leq\ & 9M\sqrt{\lceil\frac{4}{\rho^2}\log(\frac{|\mathcal{Y}|^2\rho^2 n}{4\log p})\rceil\frac{\log p}{n} + \frac{\log 2/\delta}{2n}} + \frac{2M}{\rho}\sqrt{\frac{\log p}{n}} + \frac{16M}{\rho n} \\
& + \frac{4M}{\rho}\frac{12\ln n}{\sqrt{n}}\sqrt{144(q-1)s^2\Lambda_p^2 r_q^2 \log(16s\Lambda_p r_q kn + 6k + 1)},
\end{aligned}
$$

where $s = \max_{i\in[n]}|H_i|$, $2 \leq p < \infty$, $1/p + 1/q = 1$, $r_q = \max_{i,h,y}\|\Psi_h(x_i, y)\|_q$, and $k = \sum_{i\in[n]}\sum_{h\in|H|_i}\sum_{y\in\mathcal{Y}_h}$.

That is we have

$$
R(g) - \hat{R}(\ell_{\rho,1}^{mult}(g)) \leq \mathcal{O}\left(\frac{Ms\ln n}{\rho\sqrt{n}}\left(\log^{\frac{1}{2}}(nsk)\right) + \sqrt{C(n, p, \rho, |\mathcal{Y}|, \delta)}\right),
$$

where $C(n, p, \rho, |\mathcal{Y}|, \delta) = \lceil\frac{1}{\rho^2}\log(\frac{|\mathcal{Y}|^2\rho^2 n}{4\log p})\rceil\frac{\log p}{n} + \frac{\log 2/\delta}{n}$.

By some simple transformations of notations, the conclusion of Corollary 2 in the main paper can be easily verified. The proof is over. $\qquad\square$

# D  Proof of Theorem 2

## D.1  Uniform Localized Convergence

### D.1.1  Preliminaries

Rademacher complexity is a classical tool in measuring the space complexity and can be used to bound the uniform deviation [2], however, it is worth noticing that it consider the worst-case of the element in function space, neglecting that the algorithm will likely pick functions that have a small error. [1] demonstrates that the local Rademacher complexity is more reasonable to be served as a complexity measure. Therefore, we use the local Rademahcer complexity as a tool to measure the space complexity. We introduce the following definition:

**Definition 3.** *For any $r > 0$, the local Rademacher complexity of $\mathcal{L}_\rho$ is*

$$\Re(\mathcal{L}_\rho^r) = \Re\left\{a\ell_\rho | a \in [0,1], \ell_\rho \in \mathcal{L}_\rho, R[(a\ell_\rho)^2] \leq r\right\}, \tag{11}$$

*where $R[(\ell_\rho)^2] = \mathbb{E}_{(x,y)\sim P}\left[\ell_\rho^2(x,y,f)\right]$.*

The key idea to obtain sharper generalization error bound is to choose a much smaller class $\mathcal{L}_\rho^r \in \mathcal{L}_\rho$ with as small a variance as possible, while requiring that $\ell_\rho$ is still in $\mathcal{L}_\rho^r$.

With the local Rademacher complexity, we have:

**Proposition 2.** *Assume that $\ell_\rho \in \mathcal{L}_\rho$ is bounded by $[0,M]$, where $M > 0$ is a constant. Let $r^*$ be the fixed point of $\Re(\mathcal{L}_\rho^r)$, that is $r^*$ is the solution of $\Re(\mathcal{L}_\rho^r) = r$ with respect to $r$. Then $\forall v > \max(1, \frac{\sqrt{2}}{2M})$, with probability $1 - \delta$, we have*

$$R(\ell_\rho) \leq \max\left\{\frac{v}{v-1}\hat{R}(\ell_\rho), \hat{R}(\ell_\rho) + c_M r^* + \frac{c_\delta}{n}\right\}, \tag{12}$$

*where $c_M = 18Mv$, $c_\delta = \frac{(12v+14)\log(1/\delta)}{3}$.*

### D.1.2  Proof of Proposition 2

We first prove the following three lemmas.

**Lemma 5.** *Let $\bar{\mathcal{L}}$ be the normalized loss space*

$$\bar{\mathcal{L}} = \left\{\frac{r}{\max(R(\ell_\rho^2), r)}\ell_\rho \Big| \ell_\rho \in \mathcal{L}_\rho\right\}. \tag{13}$$

*Suppose that, $\forall v > 1$,*

$$\hat{U}_n(\bar{\mathcal{L}}) := \sup_{\bar{\ell}_\rho \in \bar{\mathcal{L}}}\left\{R(\bar{\ell}_\rho) - \hat{R}(\bar{\ell}_\rho)\right\} \leq \frac{r}{Mv}.$$

*Then we have*

$$R(\ell_\rho) \leq \max\left\{\left(\frac{v}{v-1}\hat{R}(\ell_\rho)\right), \left(\hat{R}(\ell_\rho) + \frac{r}{Mv}\right)\right\}.$$

*Proof.* Note that, $\forall \bar{\ell}_\rho \in \bar{\mathcal{L}}$:

$$R(\bar{\ell}_\rho) \leq \hat{R}(\bar{\ell}_\rho) + \hat{U}_n(\bar{\mathcal{L}}) \leq \hat{R}(\bar{\ell}_\rho) + \frac{r}{Mv}. \tag{14}$$

Let us consider the two cases:

    1)  $R(\ell_\rho^2) \leq r, \ell_\rho \in \mathcal{L}_\rho$.

    2)  $R(\ell_\rho^2) > r, \ell_\rho \in \mathcal{L}_\rho$.

In the first case $\bar{\ell}_\rho = \ell_\rho$, by (14), we have

$$R(\ell_\rho) = R(\bar{\ell}_\rho) \leq \hat{R}(\bar{\ell}_\rho) + \frac{r}{Mv} = \hat{R}(\ell_\rho) + \frac{r}{Mv}. \tag{15}$$

In the second case, $\bar{\ell}_\rho = \frac{r}{R(\ell_\rho^2)} \ell_\rho$, then

$$R(\ell_\rho) - \hat{R}(\ell_\rho) \leq \hat{U}_n(\mathcal{L}_\rho) = \frac{R(\ell_\rho^2)}{r} \hat{U}_n(\bar{\mathcal{L}}) \leq \frac{M \cdot R(\ell_\rho)}{r} \frac{r}{Mv} = \frac{R(\ell_\rho)}{v}. \tag{16}$$

By combining the results of Eqs (15) and (16), the proof is over. □

**Lemma 6.** $\bar{\mathcal{L}} \subseteq \mathcal{L}_\rho^r$.

*Proof.* Let us consider $\mathcal{L}_\rho^r$ in the two cases:

1) $R(\ell_\rho^2) \leq r, \ell_\rho \in \mathcal{L}_\rho$.

2) $R(\ell_\rho^2) > r, \ell_\rho \in \mathcal{L}_\rho$.

In the first case, $\bar{\ell}_\rho = \ell_\rho$ and then:

$$R(\ell_\rho^2) = R(\bar{\ell}_\rho^2) \leq r.$$

In the second case, $R(\ell_\rho^2) > r$, so we have that

$$\bar{\ell}_\rho = \left[ \frac{r}{R(\ell_\rho^2)} \right] \ell_\rho, \ \frac{r}{R(\ell_\rho^2)} \leq 1,$$

and the following bound holds:

$$R(\bar{\ell}_\rho^2) = \left[ \frac{r}{R(\ell_\rho^2)} \right]^2 R(\ell_\rho^2) \leq \left[ \frac{r}{R(\ell_\rho^2)} \right] R(\ell_\rho^2) = r.$$

Thus, the lemma is proved. □

**Lemma 7.** $\psi_n(r) = \Re(\mathcal{L}_\rho^r)$ *is a sub-root function.*

*Proof.* In order to prove the lemma, the following properties mush apply:

1) $\psi_n(r)$ is positive

2) $\psi_n(r)$ is non-decrasing

3) $\psi_n(r)/\sqrt{r}$ is non-increasing

By the definition of $\Re(\mathcal{L}_\rho^r)$, it is easy to verity that $\Re(\mathcal{L}_\rho^r)$ is positive.

Concerning the second property, we have that, for $0 \leq r_1 \leq r_2$: $\mathcal{L}_\rho^{r_1} \subseteq \mathcal{L}_\rho^{r_2}$, therefore

$$\psi_n(r_1) = \mathbb{E}_{S,\sigma} \left[ \sup_{\ell_\rho \in \mathcal{L}_\rho^{r_1}} \left| \frac{1}{n} \sum_{i=1}^n \sigma_i \ell_\rho(x_i, y_i, f) \right| \right]$$

$$\leq \mathbb{E}_{S,\sigma} \left[ \sup_{\ell_\rho \in \mathcal{L}_\rho^{r_2}} \left| \frac{1}{n} \sum_{i=1}^n \sigma_i \ell_\rho(x_i, y_i, f) \right| \right]$$

$$= \psi_n(r_2).$$

Finally, concerning the third property, for $0 \leq r_1 \leq r_2$, let

$$\ell_\rho^{r_2} = \arg\sup_{\ell_\rho \in \mathcal{L}_\rho^{r_2}} \mathbb{E}_{S,\sigma} \left[ \sup_{\ell_\rho \in \mathcal{L}_\rho^{r_2}} \left| \frac{1}{n} \sum_{i=1}^n \sigma_i \ell_\rho(x_i, y_i, f) \right| \right].$$

Note that, since $\frac{r_1}{r_2} \leq 1$, we have that $\sqrt{\frac{r_1}{r_2}} \ell_\rho^{r_2} \in \mathcal{L}_\rho^{r_2}$. Consequently:

$$R\left[\left(\sqrt{\frac{r_1}{r_2}} \ell_\rho^{r_2}\right)\right]^2 = \frac{r_1}{r_2} R\left[(\ell_\rho^{r_2})^2\right] \leq r_1.$$

Thus, we have that:

$$\begin{aligned}
\psi_n(r_1) &= \mathbb{E}_{S,\sigma}\left[\sup_{\ell_\rho \in \mathcal{L}_\rho^{r_1}} \left|\frac{1}{n}\sum_{i=1}^n \sigma_i \ell_\rho(x_i, y_i, f)\right|\right] \\
&\geq \mathbb{E}_{S,\sigma}\left[\left|\frac{1}{n}\sum_{i=1}^n \sigma_i \sqrt{\frac{r_1}{r_2}}\ell_\rho^{r_2}(x_i, y_i, f)\right|\right] \\
&= \sqrt{\frac{r_1}{r_2}}\mathbb{E}_{S,\sigma}\left[\sup_{\ell_\rho \in \mathcal{L}_\rho^{r_2}} \left|\frac{1}{n}\sum_{i=1}^n \sigma_i \ell_\rho(x_i, y_i, f)\right|\right] \\
&= \sqrt{\frac{r_1}{r_2}}\psi_n(r_2),
\end{aligned}$$

which allows proving the claim since

$$\frac{\psi_n(r_2)}{\sqrt{r_2}} \leq \frac{\psi_n(r_1)}{\sqrt{r_1}}.$$

$\square$

*Proof of Proposition 2*. According to Theorem 2.1 of [1], we have

$$\begin{aligned}
\hat{U}_n(\bar{\mathcal{L}}) &= \sup_{\bar{\ell}_\rho \in \bar{\mathcal{L}}} \left\{R(\bar{\ell}_\rho) - \hat{R}(\bar{\ell}_\rho)\right\} \\
&\leq \inf_{\alpha > 0}\left(2(1+\alpha)\Re(\bar{\mathcal{L}}) + \sqrt{\frac{2r\log(1/\delta)}{n}}\right. \\
&\quad \left. + M\left(\frac{1}{3} + \frac{1}{\alpha}\right)\frac{\log(1/\delta)}{n}\right) \\
&\leq \inf_{\alpha > 0}\left(2(1+\alpha)\Re(\mathcal{L}_\rho^r) + \sqrt{\frac{2r\log(1/\delta)}{n}}\right. \\
&\quad \left. + M\left(\frac{1}{3} + \frac{1}{\alpha}\right)\frac{\log(1/\delta)}{n}\right) \quad \text{Using Lemma 6} \\
&\leq 3\Re(\mathcal{L}_\rho^r) + \sqrt{\frac{2r\log(1/\delta)}{n}} + \frac{7M\log(1/\delta)}{3n} \quad \text{Setting } \alpha = 1/2 \\
&\leq 3\sqrt{rr^*} + \sqrt{\frac{2r\log(1/\delta)}{n}} + \frac{7M\log(1/\delta)}{3n} \quad \text{Using sub-root property.}
\end{aligned}$$

The last step of the proof consists in showing that $r$ can be chosen, such that $\hat{U}_n(\bar{\mathcal{L}}) \leq \frac{r}{Mv}$ and $r \geq r^*$, so that we can exploit Lemma 5 and finish the proof. For this purpose, we set

$$A = 3\sqrt{r^*} + \sqrt{\frac{2\log(1/\delta)}{n}}, B = \frac{7M\log(1/\delta)}{3n}.$$

Thus, we have to find the solution of

$$A\sqrt{r} + B = \frac{r}{Mv},$$

which is

$$r = \frac{\left[\left(\frac{2B}{vM} + A^2\right) + \sqrt{\left(\frac{2B}{vM} + A^2\right)^2 - \frac{4B^2}{M^2 v^2}}\right]}{\frac{2}{M^2 v^2}} \tag{17}$$

Since $v \geq \max(1, \frac{\sqrt{2}}{2M})$, $v^2 M^2 \geq \frac{1}{2}$. Therefore, from (17), we have

$$r \geq A^2 M^2 v^2 \geq \frac{A^2}{2} = r^*,$$
$$r \leq A^2 M^2 v^2 + 2BMv.$$

Thus, we have

$$\frac{r}{Mv} \leq A^2 Mv + 2B$$
$$= \left(3\sqrt{r^*} + \sqrt{\frac{2\log(1/\delta)}{n}}\right)^2 Mv + \frac{14M\log(1/\delta)}{3n}.$$

Note that, $\forall a, b > 0$, $(a+b)^2 \leq 2a^2 + 2b^2$, so we have that

$$\frac{r}{Mv} \leq 18Mvr^* + \frac{(12v + 14)\log(1/\delta)}{3n}.$$

By substituting the above inequality into Lemma 5, the proof is over. $\qquad\square$

### D.2 Proof of Theorem 2

*Proof.* The key step is to obtain the fixed point $r^*$ of $\Re(\mathcal{L}_\rho^r)$. According to Lemma 3.6 of [11], with probability $1 - \delta$, we have

$$\Re(\mathcal{L}_\rho^r) \leq \hat{\Re}(\mathcal{L}_\rho^r) + \sqrt{\frac{2\log(1/\delta)\Re(\mathcal{L}_\rho^r)}{n}}.$$

Note that $\forall a, b > 0$, $\sqrt{ab} \leq \frac{a}{2} + \frac{b}{2}$. So we have

$$\Re(\mathcal{L}_\rho^r) \leq \hat{\Re}(\mathcal{L}_\rho^r) + \Re(\mathcal{L}_\rho^r)/2 + \frac{\log(1/\delta)}{n}.$$

There holds that

$$\Re(\mathcal{L}_\rho^r) \leq 2\hat{\Re}(\mathcal{L}_\rho^r) + \frac{2\log(1/\delta)}{n}.$$

Based on the Lemma 2.2 of [13], we have that

$$\hat{\Re}(\mathcal{L}_\rho^r) \leq 21\sqrt{6\beta r}\log^{3/2}(64n)\hat{\Re}(\widetilde{\mathcal{F}}_\rho).$$

Thus, we have

$$\Re(\mathcal{L}_\rho^r) \leq 42\sqrt{6\beta r}\log^{3/2}(64n)\hat{\Re}(\widetilde{\mathcal{F}}_\rho) + \frac{2\log(1/\delta)}{n}.$$

In the *proof of Theorem 1* part, we have

$$\hat{\Re}(\widetilde{\mathcal{F}}_\rho) \leq \frac{4}{n} + \frac{12\ln n}{\sqrt{n}}\sqrt{a\log(bn + c)},$$

where $a := 144(q-1)s^2\Lambda_p^2 r_q^2$, $b := 16s\Lambda_p r_q k$ and $c = 6k + 1$, and where $s = \max_{i\in[n]}|H_i|$ and $k = \sum_{i\in[n]}\sum_{h\in|H|_i}\sum_{y\in\mathcal{Y}_h}$.

So we obtain

$$\Re(\mathcal{L}_\rho^r) \leq 42\sqrt{6\beta r}\log^{3/2}(64n)\left[\frac{4}{n} + \frac{12\ln n}{\sqrt{n}}\sqrt{a\log(bn + c)}\right] + \frac{2\log(1/\delta)}{n}.$$

Therefore, we set

$$\psi(r) = 42\sqrt{6\beta r}\log^{3/2}(64n)\left[\frac{4}{n} + \frac{12\ln n}{\sqrt{n}}\sqrt{a\log(bn + c)}\right] + \frac{2\log(1/\delta)}{n}.$$

Solving the equation $\psi(r^*) = r^*$, we obtain

$$r^* = C \left[ \beta \frac{\log^2(n) \ln^2 n}{n} a \log(bn + c) + \frac{\log(1/\delta)}{n} \right]$$

$$\leq C \left[ \beta \frac{\log^4 n}{n} a \log(bn + c) + \frac{\log(1/\delta)}{n} \right],$$

where $C$ is a constant. This result show that $\forall v > \max(1, \frac{\sqrt{2}}{2M})$, with probability $1 - 2\delta$, we have

$$R(\ell_\rho) \leq \max \left\{ \frac{v}{v-1} \hat{R}(\ell_\rho), \hat{R}(\ell_\rho) + C \frac{\beta \log^4 n}{n} a \log(bn + c) + \frac{C \log(\frac{1}{\delta})}{n} \right\}.$$

Therefore, we have $\forall v > \max(1, \frac{\sqrt{2}}{2M})$, for any $\delta > 0$, with probability $1 - \delta$ over the sample $S$, we have

$$R(f) \leq R(\ell(\rho_f)) \leq \max \left\{ \frac{v}{v-1} \hat{R}(\ell(\rho_f)), \hat{R}(\ell(\rho_f)) + \mathcal{O} \left( \frac{\beta s^2 \log^4 n}{n} \log(nsk) + \frac{\log(\frac{1}{\delta})}{n} \right) \right\}.$$

for any $f \in \mathcal{F}$, where $s = \max_{i \in [n]} |H_i|$ and $k = \sum_{i \in [n]} \sum_{h \in |H|_i} \sum_{y \in \mathcal{Y}_h}$.

By some simple transformations of notations, the conclusion of Theorem 2 in the main paper can be easily verified. The proof is over. $\qquad \square$

**Remark 2. [Sketch of proof techniques.]** We first use uniform localized convergence technique [1] to prove Proposition 2: $\forall v > \max(1, \frac{\sqrt{2}}{2M})$, with probability $1 - \delta$, we have $R(\ell_\rho) \leq \max\{\frac{v}{v-1}\hat{R}(\ell_\rho), \hat{R}(\ell_\rho) + c_M r^* + \frac{c_\delta}{n}\}$, where $c_M$ and $c_\delta$ are constant, and $r^*$ is the solution of $\Re(\mathcal{L}_\rho^r) = r$ with respect to $r$. Thus the key step is to bound the local Rademacher complexity term $\Re(\mathcal{L}_\rho^r)$ to find its $r^*$. And to use the covering number bound of the margin function class $\log \mathcal{N}_\infty(\epsilon, \widetilde{\mathcal{F}}_\rho, S)$ established in the proof of Theorem 1, it is necessary for us to construct the relationship between the local Rademacher complexity $\Re(\mathcal{L}_\rho^r)$ of the loss function class $\mathcal{L}_\rho$ and the covering number bound of the margin function. We deal with it by using the smooth property of $\mathcal{L}_\rho$ to bound the local Rademacher complexity $\Re(\mathcal{L}_\rho^r)$ by $\mathcal{O}(\sqrt{r}\hat{\Re}(\widetilde{\mathcal{F}}_\rho))$ and by using the refined Dudley entropy integral inequality [4] with $\ell_\infty$-norm to bound $\hat{\Re}(\widetilde{\mathcal{F}}_\rho)$). Thus the relationship is built. Finally, solving $r^*$ of the upper bound established for $\Re(\mathcal{L}_\rho^r)$ finishes the proof.

## E  Proof of Theorem 3

### E.1  Preliminaries

We first introduce four Lemmas. The first Lemma is a slight refined result of Theorem 2 in [5].

**Lemma 8.** *Let $\mathcal{L}_\rho$ be a function class defined in (1) satisfying $\|\ell_\rho\|_\infty \leq M$, $\forall \ell_\rho \in \mathcal{L}_\rho$. There holds the following inequality:*

$$\Re(\mathcal{L}_\rho^r) \leq \inf_{\epsilon > 0} \left\{ 2\Re\{\ell_\rho \in \widetilde{\mathcal{L}}_\rho : \hat{R}(\ell_\rho^2) \leq \epsilon^2\} + \frac{8M\mathbb{E}\log\mathcal{N}_\infty(\epsilon/2, \mathcal{L}_\rho, S)}{n} + \sqrt{\frac{2r\mathbb{E}\log\mathcal{N}_\infty(\epsilon/2, \mathcal{L}_\rho, S)}{n}} \right\},$$

*where $\widetilde{\mathcal{L}}_\rho := \{\ell_\rho - \ell_\rho' : \ell_\rho, \ell_\rho' \in \mathcal{L}_\rho\}$, $\hat{R}(\ell_\rho^2) = \frac{1}{n} \sum_{i=1}^n \ell(\rho_f(x_i, y_i))^2$ and $R[(\ell_\rho)^2] = \mathbb{E}_{(x,y) \sim P} [\ell(\rho_f(x, y))^2]$.*

*Proof.* According to Lemma 1 in [5] and $\log\mathcal{N}_2(\epsilon/2, \mathcal{L}_\rho, S) \leq \log\mathcal{N}_\infty(\epsilon/2, \mathcal{L}_\rho, S)$, we have

$$\mathbb{E}_\sigma \hat{\Re}\{\ell_\rho \in \mathcal{L}_\rho : \hat{R}(\ell_\rho^2) \leq r) \leq \inf_{\epsilon > 0} \left\{ \mathbb{E}_\sigma \hat{\Re}\{\ell_\rho \in \widetilde{\mathcal{L}}_\rho : \hat{R}(\ell_\rho^2) \leq \epsilon^2\} + \sqrt{\frac{2r\log\mathcal{N}_\infty(\epsilon/2, \mathcal{L}_\rho, S)}{n}} \right\}. \tag{18}$$

For any $\epsilon > 0$, we fix the sample $S$. For any $\ell_\rho \in \mathcal{L}_\rho$ with $R(\ell_\rho^2) \leq r$, there holds that

$$\hat{R}(\ell_\rho^2) \leq \sup_{\ell_\rho \in \mathcal{L}_\rho : R(\ell_\rho^2) \leq r} (\hat{R}(\ell_\rho^2) - R(\ell_\rho^2)) + R(\ell_\rho^2) \leq \sup_{\ell_\rho \in \mathcal{L}_\rho : R(\ell_\rho^2) \leq r} (\hat{R}(\ell_\rho^2) - R(\ell_\rho^2)) + r.$$

Therefore, there holds almost surely that

$$\{\ell_\rho \in \mathcal{L}_\rho : R(\ell_\rho^2) \le r\} \subseteq \left\{\ell_\rho \in \mathcal{L}_\rho : \hat{R}(\ell_\rho^2) \le \sup_{\ell_\rho \in \mathcal{L}_\rho : R(\ell_\rho^2) \le r} (\hat{R}(\ell_\rho^2) - R(\ell_\rho^2)) + r\right\}.$$

This imply that

$$\Re(\mathcal{L}_\rho^r) = \mathbb{E}\mathbb{E}_{\boldsymbol{\sigma}}\hat{\Re}\{\ell_\rho \in \mathcal{L}_\rho : R(\ell_\rho^2) \le r\}$$

$$\le \mathbb{E}\mathbb{E}_{\boldsymbol{\sigma}}\hat{\Re}\left\{\ell_\rho \in \mathcal{L}_\rho : \hat{R}(\ell_\rho^2) \le \sup_{\ell_\rho \in \mathcal{L}_\rho : R(\ell_\rho^2) \le r} (\hat{R}(\ell_\rho^2) - R(\ell_\rho^2)) + r\right\}$$

$$\le \mathbb{E}\hat{\Re}\{\ell_\rho \in \widetilde{\mathcal{L}_\rho} : \hat{R}(\ell_\rho^2) \le \epsilon^2\} + \sqrt{\frac{2}{n}}\mathbb{E}\sqrt{\left(\sup_{\ell_\rho \in \mathcal{L}_\rho : R(\ell_\rho^2) \le r} (\hat{R}(\ell_\rho^2) - R(\ell_\rho^2)) + r\right)\log\mathcal{N}_\infty(\epsilon/2, \mathcal{L}_\rho, S)}$$

$$\le \mathbb{E}\hat{\Re}\{\ell_\rho \in \widetilde{\mathcal{L}_\rho} : \hat{R}(\ell_\rho^2) \le \epsilon^2\} + \sqrt{\frac{2\mathbb{E}\log\mathcal{N}_\infty(\epsilon/2, \mathcal{L}_\rho, S)}{n}}\sqrt{\left(\mathbb{E}\sup_{\ell_\rho \in \mathcal{L}_\rho : R(\ell_\rho^2) \le r} (\hat{R}(\ell_\rho^2) - R(\ell_\rho^2)) + r\right)},$$

where the second inequality follows from (18) and the last inequality follows the concavity of $f(x) = \sqrt{x}$ coupled with the Jensen's inequality. Besides, according to the standard symmetrical inequality on Rademacher average and the Lipschite property of $f(x) = x^2$ with lipschitz constant $2M$ on $[-M, M]$ (a direct application of Lemma A.4 in [5]), there holds that

$$\sqrt{\left(\mathbb{E}\sup_{\ell_\rho \in \mathcal{L}_\rho : R(\ell_\rho^2) \le r} (\hat{R}(\ell_\rho^2) - R(\ell_\rho^2)) + r\right)} \le \sqrt{2\mathbb{E}\hat{\Re}\{\ell_\rho^2 : \ell_\rho \in \mathcal{L}_\rho : R(\ell_\rho^2) \le r\} + r}$$

$$\le \sqrt{4M\mathbb{E}\hat{\Re}\{\ell_\rho \in \mathcal{L}_\rho : R(\ell_\rho^2) \le r\} + r}.$$

Thus, we obtain that

$$\Re(\mathcal{L}_\rho^r) = \mathbb{E}\mathbb{E}_{\boldsymbol{\sigma}}\hat{\Re}\{\ell_\rho \in \mathcal{L}_\rho : R(\ell_\rho^2) \le r\}$$

$$\le \mathbb{E}\hat{\Re}\{\ell_\rho \in \widetilde{\mathcal{L}_\rho} : \hat{R}(\ell_\rho^2) \le \epsilon^2\} + \sqrt{\frac{2\mathbb{E}\log\mathcal{N}_\infty(\epsilon/2, \mathcal{L}_\rho, S)}{n}}\sqrt{4M\mathbb{E}\hat{\Re}\{\ell_\rho \in \mathcal{L}_\rho : R(\ell_\rho^2) \le r\} + r}.$$

Solving this inequality, we have

$$\Re(\mathcal{L}_\rho^r) \le 2\mathbb{E}\hat{\Re}\{\ell_\rho \in \widetilde{\mathcal{L}_\rho} : \hat{R}(\ell_\rho^2) \le \epsilon^2\} + \frac{8M\mathbb{E}\log\mathcal{N}_\infty(\epsilon/2, \mathcal{L}_\rho, S)}{n} + \sqrt{\frac{2r\mathbb{E}\log\mathcal{N}_\infty(\epsilon/2, \mathcal{L}_\rho, S)}{n}}.$$

The above inequality is hold for all $\epsilon > 0$, thus the proof is over. $\qquad\square$

**Lemma 9** ([9]). *Let $S = \{X_1, ..., X_n\}$ be a set of examples and let $P_n$ be the associated empirical measure. For any function class $\mathcal{F}$ and any monotone sequence $(\epsilon_K)_{k=0}^\infty$ decreasing to $0$ such that $\epsilon_0 \ge \sup_{f \in \mathcal{F}} \sqrt{P_n f^2}$, the following inequality holds for every non-negative integer $N$:*

$$\hat{\Re}(\mathcal{F}) \le 4\sum_{k=1}^N \epsilon_{k-1}\sqrt{\frac{\log\mathcal{N}_\infty(\epsilon_k, \mathcal{F}, S)}{n}} + \epsilon_N.$$

**Lemma 10** ([12]). *Let $\|\cdot\|$ be a norm defined on the class $\mathcal{F}$. Define $\widetilde{\mathcal{F}}$ as $\{f - g : f, g \in \mathcal{F}\}$, we have $\mathcal{N}(\epsilon, \widetilde{\mathcal{F}}, \|\cdot\|) \le \mathcal{N}^2(\epsilon/2, \mathcal{F}, \|\cdot\|)$.*

**Lemma 11** ([12]). *Let $\mathcal{F}$ be a class of functions from $\mathcal{X}$ to $\mathbb{R}$ and let $\mathcal{F}_0 \subseteq \mathcal{F}$ be a subset. Then for any $\epsilon > 0$, we have the following relationship on covering number: $\mathcal{N}(\epsilon, \mathcal{F}_0, d) \le \mathcal{N}(\epsilon/2, \mathcal{F}, d)$.*

### E.2   Proof of Theorem 3

*Proof.* Recall that the loss function space is defined as:

$$\mathcal{L}_\rho = \{\ell_\rho(x, y, f) := \ell(\rho_f(x, y)) : f \in \mathcal{F}\},$$

and the margin function space

$$\widetilde{\mathcal{F}}_\rho = \{(x,y) \mapsto \rho_f(x,y) : f \in \mathcal{F}_p\}.$$

Since $\ell(\rho_f)$ is $\mu$-Lipschitz continuous w.r.t $\rho_f$, so there holds that

$$\log \mathcal{N}_\infty(\epsilon, \mathcal{L}_\rho, S) \leq \log \mathcal{N}_\infty(\epsilon/\mu, \widetilde{\mathcal{F}}_\rho, S). \tag{19}$$

Thus based on the result of Lemma 8, we have

$$\Re(\mathcal{L}_\rho^r)$$

$$\leq \inf_{\epsilon > 0} \left\{ 2\mathbb{E}\hat{\Re}\{\ell_\rho \in \widetilde{\mathcal{L}_\rho} : \hat{R}(\ell_\rho^2) \leq \epsilon^2\} + \frac{8M\mathbb{E}\log \mathcal{N}_\infty(\epsilon/2, \mathcal{L}_\rho, S)}{n} + \sqrt{\frac{2r\mathbb{E}\log \mathcal{N}_\infty(\epsilon/2, \mathcal{L}_\rho, S)}{n}} \right\}$$

$$\leq \inf_{\epsilon > 0} \left\{ 2\mathbb{E}\hat{\Re}\{\ell_\rho \in \widetilde{\mathcal{L}_\rho} : \hat{R}(\ell_\rho^2) \leq \epsilon^2\} + \frac{8M\mathbb{E}\log \mathcal{N}_\infty(\epsilon/2\mu, \widetilde{\mathcal{F}}_\rho, S)}{n} + \sqrt{\frac{2r\mathbb{E}\log \mathcal{N}_\infty(\epsilon/2\mu, \widetilde{\mathcal{F}}_\rho, S)}{n}} \right\}.$$

For the term $\mathbb{E}\hat{\Re}\{\ell_\rho \in \widetilde{\mathcal{L}_\rho} : \hat{R}(\ell_\rho^2) \leq \epsilon^2\}$, applying Lemma 9 with the assignment $\epsilon_k = 2^{-k}\epsilon$, we have

$$\mathbb{E}\hat{\Re}\{\ell_\rho \in \widetilde{\mathcal{L}_\rho} : \hat{R}(\ell_\rho^2) \leq \epsilon^2\} \leq 4\mathbb{E}\sum_{k=1}^{N} \epsilon_{k-1}\sqrt{\frac{\log \mathcal{N}_\infty(\epsilon_k/2, \widetilde{\mathcal{L}_\rho}, S)}{n}} + \epsilon_N$$

$$\leq 4\mathbb{E}\sum_{k=1}^{N} \epsilon_{k-1}\sqrt{\frac{2\log \mathcal{N}_\infty(\epsilon_k/4, \mathcal{L}_\rho, S)}{n}} + \epsilon_N$$

$$\leq 4\mathbb{E}\sum_{k=1}^{N} \epsilon_{k-1}\sqrt{\frac{2\log \mathcal{N}_\infty(\epsilon_k/4\mu, \widetilde{\mathcal{F}}_\rho, S)}{n}} + \epsilon_N,$$

where the first inequality follows from Lemma 11, the second inequality follows from Lemma 10, the third inequality follows from (19).

Thus, the important term need to bound is $\log \mathcal{N}_\infty(\widetilde{\mathcal{F}}_\rho, \epsilon, S)$. As we showed in (8) of Proposition 1, there holds that

$$\log \mathcal{N}_\infty(\epsilon, \widetilde{\mathcal{F}}_\rho, S) \leq \log \mathcal{N}_\infty(\frac{1}{2s}\epsilon, \widetilde{F}, \widetilde{S}),$$

where

$$\widetilde{F} := \{v \mapsto \langle w, v \rangle : w \in \mathbb{R}^N, \|w\|_p \leq \Lambda_p, v \in \widetilde{S}\},$$

and where $\widetilde{S}$ is defined as follows

$$\widetilde{S} := \{\Psi_h(x,y) : x \in \{x_1, ..., x_n\}, h \in H_i, y \in \mathcal{Y}_h\}.$$

(1.) According to the Assumption 3 of the main paper, we have

$$\log \mathcal{N}_\infty(\epsilon, \widetilde{F}, \widetilde{S}) \leq \frac{\gamma_p}{\epsilon^p}.$$

Then we have

$$\log \mathcal{N}_\infty(\epsilon, \widetilde{\mathcal{F}}_\rho, S) \leq \frac{2^p s^p \gamma_p}{\epsilon^p}.$$

Based on the above analysis and results, we have

$$\mathbb{E}\hat{\Re}\{\ell_\rho \in \widetilde{\mathcal{L}_\rho} : \hat{R}(\ell_\rho^2) \leq \epsilon^2\} \leq 4\mathbb{E}\sum_{k=1}^{N} \epsilon_{k-1}\sqrt{\frac{2\log \mathcal{N}_\infty(\epsilon_k/4\mu, \widetilde{\mathcal{F}}_\rho, S)}{n}} + \epsilon_N$$

$$\leq 4\sum_{k=1}^{N} 2^{1-k}\epsilon\sqrt{\frac{2^{3p+kp+1}\gamma_p \mu^p s^p \epsilon^{-p}}{n}} + 2^{-N}\epsilon$$

$$= \sqrt{\frac{\gamma_p \mu^p s^p}{n}} 2^{(3p+7)/2}\epsilon^{1-p/2}\sum_{k=1}^{N} 2^{(p-2)k/2} + 2^{-N}\epsilon,$$

when $0 < p < 2$, the series $\sum_{k=1}^{+\infty} 2^{(p-2)k/2}$ converges and thus one can tend $N \to \infty$ to derive the bound $\mathbb{E}\hat{\Re}\{\ell_\rho \in \widetilde{\mathcal{L}_\rho} : \hat{R}(\ell_\rho^2) \le \epsilon^2\} \le c_p\sqrt{\frac{\gamma_p \mu^p s^p}{n}}\epsilon^{1-p/2}$. Therefore, we obtain that

$$\Re(\mathcal{L}_\rho^r) \le c_p \inf_{\epsilon>0} \left\{ \sqrt{\frac{\gamma_p \mu^p s^p}{n}}\epsilon^{1-p/2} + \frac{8M\mathbb{E}\log\mathcal{N}_\infty(\epsilon/2\mu, \widetilde{\mathcal{F}}_\rho, S)}{n} + \sqrt{\frac{2r\mathbb{E}\log\mathcal{N}_\infty(\epsilon/2\mu, \widetilde{\mathcal{F}}_\rho, S)}{n}} \right\}$$

$$\le c_p \inf_{\epsilon>0} \left\{ \sqrt{\frac{\gamma_p \mu^p s^p}{n}}\epsilon^{1-p/2} + \frac{8M2^{2p}\gamma_p \mu^p s^p}{n\epsilon^p} + \sqrt{\frac{2r2^{2p}\gamma_p \mu^p s^p}{n\epsilon^p}} \right\}$$

$$\le c_{p,M} \inf_{\epsilon>0} \left\{ \sqrt{\frac{\gamma_p \mu^p s^p}{n}}\epsilon^{1-p/2} + \frac{\gamma_p \mu^p s^p}{n\epsilon^p} + \sqrt{\frac{r\gamma_p \mu^p s^p}{n\epsilon^p}} \right\}.$$

So we obtain that

$$\psi_\epsilon(r) = c_{p,M} \inf_{\epsilon>0} \left\{ \sqrt{\frac{\gamma_p \mu^p s^p}{n}}\epsilon^{1-p/2} + \frac{\gamma_p \mu^p s^p}{n\epsilon^p} + \sqrt{\frac{r\gamma_p \mu^p s^p}{n\epsilon^p}} \right\}.$$

The associated fixed point $r_\epsilon^* = \psi_\epsilon(r_\epsilon^*)$ satisfies the constraint

$$r_\epsilon^* \le c_{p,M} \left[ \sqrt{\frac{\gamma_p \mu^p s^p}{n}}\epsilon^{1-p/2} + \frac{\gamma_p \mu^p s^p}{n\epsilon^p} \right].$$

If we choose $\epsilon = n^{-1/(p+2)}$, we obtain

$$r_\epsilon^* = c_{p,M} \frac{\gamma_p \mu^p s^p}{n^{\frac{2}{p+2}}}.$$

This result show that $\forall v > \max(1, \frac{\sqrt{2}}{2M})$, with probability $1 - \delta$, we have

$$R(\ell_\rho) \le \max\left\{ \frac{v}{v-1}\hat{R}(\ell_\rho), \hat{R}(\ell_\rho) + c_{p,M}\frac{\gamma_p \mu^p s^p}{n^{\frac{2}{p+2}}} + \frac{c_\delta}{n} \right\}.$$

That is $\forall v > \max(1, \frac{\sqrt{2}}{2M})$, for any $\delta > 0$, with probability $1 - \delta$ over the sample $S$, we have

$$R(f) \le R(\ell(\rho_f)) \le \max\left\{ \frac{v}{v-1}\hat{R}(\ell(\rho_f)), \hat{R}(\ell(\rho_f)) + \mathcal{O}\left( \frac{\gamma_p \mu^p s^p}{n^{\frac{2}{p+2}}} + \frac{\log(\frac{1}{\delta})}{n} \right) \right\}.$$

for any $f \in \mathcal{F}$, where $0 < p < 2$ and $s = \max_{i\in[n]} |H_i|$.

**(2.)** Similarly, according to the Assumption 4 of the main paper, we have

$$\log\mathcal{N}_\infty(\epsilon, \widetilde{F}, \widetilde{S}) \le D\log^p(\frac{\gamma}{\epsilon}).$$

Then we have

$$\log\mathcal{N}_\infty(\epsilon, \widetilde{\mathcal{F}}_\rho, S) \le D\log^p(\frac{2s\gamma}{\epsilon}).$$

Based on the above analysis and results, we have the following inequality that holds for any $N \in \mathbb{N}^+$:

$$\mathbb{E}\hat{\Re}\{\ell_\rho \in \widetilde{\mathcal{L}_\rho} : \hat{R}(\ell_\rho^2) \le \epsilon^2\} \le 4\mathbb{E}\sum_{k=1}^N \epsilon_{k-1}\sqrt{\frac{2\log\mathcal{N}_\infty(\epsilon_k/4\mu, \widetilde{\mathcal{F}}_\rho, S)}{n}} + \epsilon_N$$

$$\le 4\sum_{k=1}^N 2^{1-k}\epsilon\sqrt{\frac{2D\log^p(\frac{8\mu s\gamma}{\epsilon_k})}{n}} + \epsilon_N$$

$$\le 2^{7/2}\sqrt{\frac{D}{n}}\sum_{k=1}^N 2^{-k}\epsilon\log^{p/2}(2^{k+3}\mu s\gamma\epsilon^{-1}) + \epsilon_N$$

$$\le 2^{(7+p)/2}\sqrt{\frac{D}{n}}\sum_{k=1}^N 2^{-k}\epsilon[((k+1)\log 2)^{p/2} + \log^{p/2}(4\mu s\gamma\epsilon^{-1})] + \epsilon_N$$

$$\le 2^{(7+p)/2}\sqrt{\frac{D}{n}}\epsilon[c(p) + \log^{p/2}(4\mu s\gamma\epsilon^{-1})] + \epsilon_N,$$

$$(20)$$

where the fourth inequality follows from $(a+b)^{p/2} \leq [2\max(a,b)]^{p/2} \leq 2^{p/2}(a^{p/2}+b^{p/2}), a, b \geq 0$ and the last inequality is due to the fact $\sum_{k=1}^{N} 2^{-k}((k+2)\log 2)^{p/2} < \infty$.
Letting $N \to \infty$ in (20), we have

$$\Re(\mathcal{L}_\rho^r) \leq \inf_{\epsilon>0} \left\{ 2^{(9+p)/2}\sqrt{\frac{D}{n}}\epsilon[c(p) + \log^{p/2}(4\mu s\gamma\epsilon^{-1})] + \frac{8M\mathbb{E}\log\mathcal{N}_\infty(\epsilon/2\mu, \widetilde{\mathcal{F}}_\rho, S)}{n} \right.$$

$$\left. + \sqrt{\frac{2r\mathbb{E}\log\mathcal{N}_\infty(\epsilon/2\mu, \widetilde{\mathcal{F}}_\rho, S)}{n}} \right\}$$

$$\leq \inf_{\epsilon>0} \left\{ 2^{(9+p)/2}\sqrt{\frac{D}{n}}\epsilon[c(p) + \log^{p/2}(4\mu s\gamma\epsilon^{-1})] + \frac{8MD\log^p(\frac{4\mu s\gamma}{\epsilon})}{n} + \sqrt{\frac{2rD\log^p(\frac{4\mu s\gamma}{\epsilon})}{n}} \right\}$$

$$\leq c\inf_{\epsilon>0} \left\{ \sqrt{\frac{D}{n}}\epsilon\log^{p/2}(4\mu s\gamma\epsilon^{-1}) + \frac{D\log^p(\frac{4\mu s\gamma}{\epsilon})}{n} + \sqrt{\frac{rD\log^p(\frac{4\mu s\gamma}{\epsilon})}{n}} \right\}.$$
(21)

Setting $\epsilon = \sqrt{r}$ in (21), we obtain that

$$\Re(\mathcal{L}_\rho^r) \leq c\left[ \frac{D\log^p(\frac{4\mu s\gamma}{\sqrt{r}})}{n} + \sqrt{\frac{rD\log^p(\frac{4\mu s\gamma}{\sqrt{r}})}{n}} \right].$$

Setting $\epsilon = \frac{1}{\sqrt{n}}$, we derive that

$$\Re(\mathcal{L}_\rho^r) \leq c\left[ \frac{D\log^p(4\mu s\gamma\sqrt{n})}{n} + \sqrt{\frac{rD\log^p(4\mu s\gamma\sqrt{n})}{n}} \right].$$

Therefore, we obtain that

$$\psi(r) = c\left[ \frac{D\log^p(4\mu s\gamma\sqrt{n})}{n} + \sqrt{\frac{rD\log^p(4\mu s\gamma\sqrt{n})}{n}} \right].$$

The associated fixed point $r^* = \psi(r^*)$ satisfies

$$r^* = c\left[ \frac{D\log^p(4\mu s\gamma\sqrt{n})}{n} + \sqrt{\frac{r^*D\log^p(4\mu s\gamma\sqrt{n})}{n}} \right].$$

Solving this equation, we obtain

$$r^* \leq c\frac{D\log^p(\mu s\gamma\sqrt{n})}{n}.$$

This result show that $\forall v > \max(1, \frac{\sqrt{2}}{2M})$, with probability $1 - \delta$, we have

$$R(\ell_\rho) \leq \max\left\{ \frac{v}{v-1}\hat{R}(\ell_\rho), \hat{R}(\ell_\rho) + c\frac{D\log^p(\mu s\gamma\sqrt{n})}{n} + \frac{c_\delta}{n} \right\}.$$

That is $\forall v > \max(1, \frac{\sqrt{2}}{2M})$, for any $\delta > 0$, with probability $1 - \delta$ over the sample $S$, we have

$$R(f) \leq R(\ell(\rho_f)) \leq \max\left\{ \frac{v}{v-1}\hat{R}(\ell(\rho_f)), \hat{R}(\ell(\rho_f)) + \mathcal{O}\left( \frac{D\log^p(\mu s\gamma\sqrt{n})}{n} + \frac{\log(\frac{1}{\delta})}{n} \right) \right\}.$$

for any $f \in \mathcal{F}$, where $s = \max_{i\in[n]}|H_i|$. The proof is over. $\qquad\square$

**Remark 3. [Sketch of proof techniques.]** The proof is also based on Proposition 2. Thus the key step is to bound the local Rademacher complexity of the loss function class $\Re(\mathcal{L}_\rho^r)$ to find its $r^*$. In the proof of Theorem 3, the difficulty lies in constructing the inequality between $\Re(\mathcal{L}_\rho^r)$ and the covering number $\log\mathcal{N}(\epsilon, \mathcal{F}_h, \|\cdot\|_\infty)$. We handle it by slightly refining the main Theorem in [4]. Main proof techniques contain constructing the covering number inequalities among different spaces. Combined the Lipschitz property and the proof techniques in Theorem 1, we will finally need to bound the term $\log\mathcal{N}(\epsilon, \mathcal{F}_h, \|\cdot\|_\infty)$. Using Assumptions 3 or 4 and solving $r^*$ of the upper bound established for $\Re(\mathcal{L}_\rho^r)$ finish the proof.

# F    Proof of Corollary 3

*Proof.* We define the function class $\{\ell(\rho_f) - \ell(\rho_{f^*})\}$. Since $R(\ell(\rho_f) - \ell(\rho_{f^*})) \geq 0$ and $R(\ell(\rho_f) - \ell(\rho_{f^*}))^2 \leq BR(\ell(\rho_f) - \ell(\rho_{f^*}))$, if we apply the class $\{\ell(\rho_f) - \ell(\rho_{f^*})\}$ to Proposition 2, we will get

$$R(\ell(\rho_f) - \ell(\rho_{f^*})) \leq \max\left\{ \frac{v}{v-1}\left[\hat{R}(\ell(\rho_f) - \ell(\rho_{f^*}))\right], \hat{R}(\ell(\rho_f) - \ell(\rho_{f^*})) + c_B r^* + \frac{c_\delta}{n} \right\},$$

where $r^*$ is the fixed point of local Rademacher complexity of function class $\{\ell(\rho_f) - \ell(\rho_{f^*})\}$ and $c_B = 18Bv$. Note that $\hat{R}(\ell(\rho_{\hat{f}^*})) - \hat{R}(\ell(\rho_{f^*})) \leq 0$, so we have

$$R(\ell(\rho_{\hat{f}^*})) - R(\ell(\rho_{f^*})) \leq c_B r^* + \frac{c_\delta}{n}.$$

Thus the key step is to bound the local Rademacher complexity term of the function class $\{\ell(\rho_f) - \ell(\rho_{f^*})\}$ to find its $r^*$.

Recall that when we want to bound the local Rademacher complexity term of the function class $\{\ell(\rho_f)\}$ (that is $\mathcal{L}_\rho$), we can bound $\Re(\mathcal{L}_\rho^r)$ by

$$\inf_{\epsilon>0}\left\{ 2\mathbb{E}\hat{\Re}\{\ell_\rho \in \widetilde{\mathcal{L}_\rho} : \hat{R}(\ell_\rho^2) \leq \epsilon^2\} + \frac{8M\mathbb{E}\log\mathcal{N}_\infty(\epsilon/2, \mathcal{L}_\rho, S)}{n} + \sqrt{\frac{2r\mathbb{E}\log\mathcal{N}_\infty(\epsilon/2, \mathcal{L}_\rho, S)}{n}} \right\}.$$

Note that there is no difference between the metric entropy of the excess loss class $\{\ell(\rho_f) - \ell(\rho_{f^*})\}$ and mertic entropy of the loss class $\{\ell(\rho_f)\}$ itself: that is, from the definition of covering number, one has

$$\log\mathcal{N}_\infty(\epsilon, \mathcal{L}_\rho, S) = \log\mathcal{N}_\infty(\epsilon, \{\ell(\rho_f) - \ell(\rho_{f^*})\}, S).$$

Therefore, we can also bound the local Rademacher complexity of the excess loss class $\{\ell(\rho_f) - \ell(\rho_{f^*})\}$ by the following term:

$$\inf_{\epsilon>0}\left\{ 2\mathbb{E}\hat{\Re}\{\ell_\rho \in \widetilde{\mathcal{L}_\rho} : \hat{R}(\ell_\rho^2) \leq \epsilon^2\} + \frac{16M\mathbb{E}\log\mathcal{N}_\infty(\epsilon/2, \mathcal{L}_\rho, S)}{n} + \sqrt{\frac{2r\mathbb{E}\log\mathcal{N}_\infty(\epsilon/2, \mathcal{L}_\rho, S)}{n}} \right\}.$$

This means that for the local Rademacher complexity of the excess loss class $\{\ell(\rho_f) - \ell(\rho_{f^*})\}$, we finally obtain the same $r^*$ as in Theorem 3.

Therefore, under Assumptions 1 and 3 of the main paper, for any $\delta > 0$, with probability $1 - \delta$ over the sample $S$, there holds

$$R(\ell(\rho_{\hat{f}^*})) \leq R(\ell(\rho_{f^*})) + \mathcal{O}\left(\frac{\gamma_p \mu^p s^p}{n^{\frac{2}{p+2}}}\right) + \mathcal{O}\left(\frac{\log(\frac{1}{\delta})}{n}\right).$$

for any $f \in \mathcal{F}$, where $0 < p < 2$ and $s = \max_{i \in [n]} |H_i|$.

The proof for the Assumptions 1 and 4 case is similar. The proof is over.    □

**Remark 4.** We now illustrate the difference between our bound in the space capacity setting and the empirical Bernstein bound in [7]. Theorem 6 in [7] shows the following generalization bound: $R(f) - \hat{R}(f) \leq \widetilde{\mathcal{O}}(\sqrt{\frac{Var_n(f,S)\ln(\mathcal{N}_\infty(1/n,\mathcal{F},2n)/\delta)}{n}} + \frac{\ln(\mathcal{N}_\infty(1/n,\mathcal{F},2n)/\delta)}{n})$, where $\mathcal{F}$ is the loss function class. If the variance of the loss and the covering number of the function class $\mathcal{F}$ are small, this generalization bound scale as $\widetilde{\mathcal{O}}\left(\frac{1}{n}\right)$. To explore different learning rates of structured prediction under different conditions, for instance, the smoothness curvature condition and the space capacity condition, instead of assuming directly that the variance of the loss is small, we exploit Theorem 2.1 in [2] and the property of sub-root functions to transform an upper bound with the variance to the bound with a fixed point of the local Rademacher complexity, please refer to the proof of Proposition 2. Moreover, assuming directly the covering number on the function class $\mathcal{F}$ will ignore the factor graph property of structured prediction since the factor graph is reflected in the scoring function, not the loss function. In the proof involving covering numbers, we exploit the covering number to bound the local Rademacher complexity and construct relationships of covering numbers among different function classes (please refer to the proof of Proposition 1 and Theorem 3), which thus permit us to show the explicit dependency on the properties of the factor graph and the dependency on the number of possible labels. Therefore, our proof and Theorem 6 in [7] all require the variance and the covering number to be small to obtain sharper generalization bounds. However, for the complex structured prediction problem, it requires fined analysis. We thus exploit different implementation modes.