# OpenReview forum: "Towards Sharper Generalization Bounds for Structured Prediction"
_NeurIPS.cc/2021/Conference — NeurIPS 2021 Poster_

### Official Review · Reviewer_Mdym · 2021-07-16

**Rating:** 7
**Confidence:** 3

**Summary:**

The paper investigates the generalization performance of structured prediction learning where the scoring function can be formulated via the notation of factor graphs. In three separate settings, the work derives state-of-the-art generalization bounds with fast convergence rates and weak dependency on the number of possible labels.


**Main Review:**


**Originality**:

- The paper derives state-of-the-art generalization bounds with different mild conditions imposed on the loss and the capacity of the hypothesis space. The results are general and seem to be applicable to various learning contexts.

- On the other hand, it's hard to pinpoint which technical contributions of the work are instrumental to the derivation of the results:
   - The most novel technique seems to appear in the proof of Proposition 1 that bounds the covering number of the function class $\mathcal{F}_{\rho}$, where the nonlinearity of the margin are bypassed by an intermediate linear function class. However, that is also a problem for multiclass learning, for which comparable bounds have been obtained. Is this a new technique that pertains to structured prediction?
   - For the rest of the proof, aspects that are unique to structured prediction, such as factor graph, rarely appears. It is worth highlighting such contributions in the discussions (more on this below).


**Quality and Clarity**:

- In general, the paper is well-written and well structured. The work seems to be built upon established literature and the review of related works is informative. The ideas are presented intuitively and the proofs are rigorous.

- While the results of the work are very impressive, the paper would be greatly strengthened by more discussions on the relevant context:
   - What are the main difficulties in obtaining those bounds and what analytical tools were constructed to address the issues?
   - What are the distinct characterization that set structured prediction apart from other contexts with similar bounds (such as multiclass classification) that requires significant technical adjustments ?

- The paragraph "Margin and loss function class" contains some typos and misuse of notation: "surogate"; $\ell_\rho(f, x, y)$ vs $\ell_p(x, y, f)$; $\rho_{f(x, y)}$ vs $\rho_{f}(x, y)$; the mathematical expression after the introduction of the surrogate loss (follow "that is") is incomprehensible


**Significance**:

- The problem studied in this paper is important, and the derived results advance the state of the art in a demonstrable way.
- Regardless of technical novelties, the paper lays out a complete framework that others can rely on for future works in this direction.


**Time Spent Reviewing:**

3

---

> ### Author Response · Authors · 2021-08-10
> **Response to Reviewer Mdym**
>
> **We sincerely appreciate your invaluable and constructive comments.**
>
> **R: It's hard to pinpoint which technical contributions of the work are instrumental to the derivation of the results: 1. The most novel technique seems to appear in the proof of Proposition 1 that bounds the covering number of the function class , where the nonlinearity of the margin are bypassed by an intermediate linear function class. However, that is also a problem for multiclass learning, for which comparable bounds have been obtained. Is this a new technique that pertains to structured prediction?
> 2. For the rest of the proof, aspects that are unique to structured prediction, such as factor graph, rarely appears. It is worth highlighting such contributions in the discussions (more on this below).**
>
> **A**:
> Most of the related work in multi-class classification studies generalization in the way of multi-output learning [1-6], building their analysis on the equivalent representation of the scoring function, that is $(f_1,f_2,...,f_c)$ with $f_j(x) = f(x,j)$, which is a vector-valued function and where $c$ is the number of classes. Since classical Rademacher complexity is defined on the real-valued function, thus, related work in multi-class classification often constructs a contraction lemma to bypass this difficulty. For instance, [1, 4, 5] use the Gaussian complexity to establish the contraction lemma, [2, 3] use the covering number, and [6] use the property of the sub-Gaussian random variable. While in structured prediction, when the analysis is based on the factor graph approach, the difficulties are that the factor graph has pretty complex substructures and that different examples are allowed to have different factor graphs, which makes the analysis of structured prediction hard. Therefore, instead of building a contraction lemma, we directly analyze the margin function, however, which is non-linear and hard to handle. We bypass this difficulty by constructing a simpler linear function space and use the $\ell_{\infty}$-norm covering number to build the relationship between the origin margin function space and our defined linear function class. We emphasize that the maximum operator in the margin function, the decomposition property of the scoring function, and Lemma 1 in Appendix also play a key role in the proof (refer to the proof of Proposition 1). Moreover, constructing this linear function class for structured prediction need fined analysis since factor graphs of different examples are different and complex. Then, please permit us to explain the benefit of analysis built upon the margin function. Since the property of the factor graph is mainly reflected in the scoring function, and since the margin function is a functional of the scoring function, when the covering number bound of the margin function is established, one can easily use it to derive the generalization bound if the assumptions are assumed on the margin function. In other words, we encapsulate the proof about the factor graph. This is the reason why proof about the factor graph rarely appears in the rest of the proof. Thank you very much for this review. We will add a more detailed discussion in the new version.
>
> [1] Y. Lei, Ü. Dogan, D.-X. Zhou, and M. Kloft. Data-dependent generalization bounds for multi-class classification. IEEE Transactions on Information Theory, 65(5):2995–3021, 2019.
>
> [2] H. W. Reeve and A. Kaban. Optimistic bounds for multi-output prediction. In International Conference on Machine Learning, 2020.
>
> [3] Foster, Dylan J., and Alexander Rakhlin. $\ell_{\infty}$ Vector Contraction for Rademacher Complexity. ArXiv Preprint ArXiv:1911.06468, 2019.
>
> [4] J. Li, Y. Liu, R. Yin, H. Zhang, L. Ding, and W. Wang. Multi-class learning: From theory to algorithm. In Advances in Neural Information Processing Systems, 2018.
>
> [5] Y. Lei, Ü. Dogan, A. Binder, and M. Kloft. Multi-class svms: From tighter data-dependent generalization bounds to novel algorithms. In Advances in Neural Information Processing Systems, 2015.
>
> [6] Maurer, Andreas. A Vector-Contraction Inequality for Rademacher Complexities. In Conference on Algorithmic Learning Theory, 2016.
>
> **R: What are the main difficulties in obtaining those bounds and what analytical tools were constructed to address the issues?**
>
> **A**: (1) In the proof of Theorem 1, the difficulty lies in obtaining the covering number bound of the margin function class with a better dependency on the size of the output space. To deal with the nonlinear margin function class, we construct a simpler linear function class. We leverage the fact that the margin function involves the maximum operator, the $\ell_{\infty}$-norm covering number, and Lemma 1 in Appendix to prove the covering number bound.
> (2) In the proof of Theorem 2, the difficulty lies in constructing the relationship between the local Rademacher complexity of the loss function class $\mathcal{L}\_{\rho}$ and the covering number bound of the margin function. We deal with it by using the smooth property of $\mathcal{L}\_{\rho}$ to bound the local Rademacher complexity by $\mathcal{O}(\sqrt{r} \hat{\Re}(\widetilde{\mathcal{F}}\_{\rho})) $ and by using the refined Dudley entropy integral inequality (Lemma 3 in Appendix) with $\ell\_{\infty}$-norm and Proposition 1 to bound $\hat{\Re}(\widetilde{\mathcal{F}}\_{\rho}))$. The relationship is therefore built. (3) In the proof of Theorem 3, the difficulty lies in constructing the inequality between $\Re(\mathcal{L}\_{\rho}^r)$ and the covering number $ \log \mathcal{N} (\epsilon,\mathcal{F}\_h, || \cdot ||\_{\infty} )$. We handle it by refining the main Theorem in Lei et al. (2016). Main proof techniques also contain constructing the covering number inequalities among different function spaces. Thank you very much for this review, we will add a more detailed discussion in the new version.
>
> **R: What are the distinct characterization that set structured prediction apart from other contexts with similar bounds (such as multiclass classification) that requires significant technical adjustments?**
>
> **A**: We think that the distinct characterization is the scoring function. In structured prediction, the factor graph has complex substructures and different examples are allowed to have different factor graphs, so the scoring function can not be easily modeled as multi-output learning as in multi-class classification learning. The detailed discussion refers to the first answer.
>
>
> **R: The paragraph "Margin and loss function class" contains some typos and misuse of notation: "surogate";** $\ell\_{\rho} (x,y,f)$ vs $\ell\_{\rho} (f,x,y)$; $\rho\_{f(x,y)}$ vs $\rho\_{f}(x,y)$ ; **the mathematical expression after the introduction of the surrogate loss (follow "that is") is incomprehensible.**
>
> **A**: We will carefully proofread the paper to eliminate all typos. About "that is $ 0 \leq L(\hat{f}(x), y) = L(\hat{f}(x), y)\_{\rho\_{f(x,y)} \leq 0} \leq \ell (\rho\_f(x,y)) \leq M$", the first inequality is because loss $L$ is positive, the first equality means the cost will only occur when the margin function value is negative, the second inequality is because the surrogate loss $\ell$ is larger than $L$, and the last inequality is because loss $\ell$ is bounded by $M$.

---

### Official Review · Reviewer_rCxV · 2021-07-19

**Rating:** 5
**Confidence:** 4

**Summary:**

Authors study generalization in the context of structured prediction based on factor graphs. Their goal is to improve the rates of existing generalization bounds, among their contributions we have:

1. Under Lipschitz continuity, an improvement from square root to logarithmic dependence on the size of the output space.
2. Under a smoothness condition, a faster convergence rate of $O(1/n)$, where $n$ is the sample size.
3. Under covering number assumptions, authors show a bound with no dependency on the output space size, simultaneously with faster convergence rates than $O(1/\sqrt{n})$.

**Limitations And Societal Impact:**

No.

**Main Review:**

Here I present my review along the axes from the NeurIPS reviewing guidelines:

**Originality**

- Are the tasks or methods new?
  - The generalization bounds are new.

- Is the work a novel combination of well-known techniques?
  - Yes. The work applies ideas from concentration inequalities, Rademacher complexity, and covering numbers, to name a few.

- Is it clear how this work differs from previous contributions?
  - Yes.

- Is related work adequately cited?
  - Mostly yes; however, there are some literature missing. For instance, upper bounds from London et al. (2016), in the context of stability, derive bounds that also depend on the size of each example.


**Quality**

- Is the submission technically sound?
  - I have not checked the proofs in detail but they seem sound.

- Are claims well supported (e.g., by theoretical analysis or experimental results)?
  - No experiments are provided.

- Are the methods used appropriate?
  - Yes, the theoretical tools used in the proofs are adequate for the analyses.

- Is this a complete piece of work or work in progress?
  - Complete work.

- Are the authors careful and honest about evaluating both the strengths and weaknesses of their work?
  - I was not able to see a discussion on the limitations of their analysis.


**Clarity**

- Is the submission clearly written?
  - No. See below.

- Is it well organized?
  - Yes.

- Does it adequately inform the reader?
  - N/A.


**Significance**

- Are the results important?
  - If proofs are correct, yes. These results are of interest to the learning theory community.

- Are others (researchers or practitioners) likely to use the ideas or build on them?
  - Perhaps. Theorists might be interested on improving the bounds or provide milder assumptions.

- Does the submission address a difficult task in a better way than previous work?
  - N/A. No algorithm is presented.

- Does it advance the state of the art in a demonstrable way?
  - The results presented improve the rates existing upper bounds, although I am not completely sure if the bounds are fully comparable to existing ones given the set of assumptions.

- Does it provide unique data, unique conclusions about existing data, or a unique theoretical or experimental approach?
  - N/A.

# Questions / Comments

I have mixed feelings about this manuscript. I believe the results (if correct) are worthy of publication. However, there are *a lot* of mistakes in the writing and, thus, the paper requires an extensive editing before publication. For that reason, the current state of the manuscript does not meet the bar to be published at NeurIPS.

I will comment on some issues I found in the *introduction* as it is just unrealistic to list all existing mistakes.

1. In the first sentence: CV, NLP, CB are not problems but *fields*.
2. L19: compute vision -> computer vision
3. L19: "..., protein..." -> "..., and protein..."
4. L21: "to above tasks" -> "to the above tasks"
5. L22: "that the natural" -> "that a natural". Otherwise, what is that "natural loss"?
6. L28: "Compared with" -> "Compared to". The point is to contrast, right?
7. L31: "bound studying" is confusing.
8. L33: "consider the specific algorithm and the simple" -> replace both "the" with "an" and "a", respectively.
9. L34: "more work considers". Seems odd, "other works consider"?
10. L35: "general loss." -> "general losses" or "a general loss."
11. L40: "general loss based" -> "general losses based"
12. The sentence from L41 to L43 is very disconnected. There is no a connection word to the previous sentences, and also it has an error at L41 "there has".
13. L44: using "these" is also odd. Better to use "the works above" or "the aforementioned works"
14. The use of [I.], [II.], [III.] looks very odd as well. There is an "enumerate" environment in Latex for that, roman listing is available as well.
15. L48: "..., [40] of" -> "..., and [40] of"
16. L50: "there raises a question" is also incorrect.
17. L51: "achieves" -> "achieve"
18. L52: Can the number of labels be infinite? It is typically exponential on the input size. What is an example of infinite number of *discrete* labels?
19. L54: "can provide the explicit" sounds odd.
20. L56: "shed" -> "sheds" as it refers to the approach (singular).
21. L63: "more difficult and complex", what is the difference between difficult and complex?
22. L64: "problem" -> "problems"
23. L64: "wonder to know". I am not sure if that phrase even exists.

I have listed several potential errors *just* in the introduction. I cannot even mention the ones throughout the rest of the manuscript.

---
I thank the authors for their response. I am increasing my score to 5 as I believe the technical contributions are relevant. However, the manuscript requires a serious revision on the writing, thus, I cannot increase my score further.

**Time Spent Reviewing:**

2

---

> ### Author Response · Authors · 2021-08-10
> **Response to Reviewer rCxV**
>
> **We sincerely appreciate your invaluable and constructive comments.**
>
> **R: There are a lot of mistakes in the writing and, thus, the paper requires an extensive editing before publication.**
>
> **A**: We really appreciate your patience in pointing out these typographical and grammatical errors for us. We spent a lot of time in the conclusions and proofs, ignoring the writing aspect, and we are very, very sorry for that. We assure you that we will do our best to proofread these typographical and grammatical errors. We will invite native English speakers to make careful corrections line by line to ensure that these errors are avoided in the final version. Please trust us.
>
> **R: There are some literature missing. For instance, upper bounds from London et al. (2016).**
>
> **A**: [1], [2], and [3] used the stability tool to investigate the generalization bounds of structure prediction, and provided interesting theoretical insights that structure prediction is possible to generalize from a few large examples, potentially even just one. For the number of example $n$ and the size of each example $m$, under suitable assumptions and hidden logarithmic terms, the latest paper [3] provided the generalization bounds of $\mathcal{O}(1/\sqrt{nm})$ by the stability and PAC-Bayes approach. The research ideas and proof techniques of the three papers and ours are very different. Compared with this bound, our generalization bounds decrease faster on $n$, please refer to Theorems 2 and 3 in the main paper. If we consider the case of $n>m$, our bounds are tighter and have a faster convergence rate. In addition, our results are based on the factor graph decomposition approach, which is beneficial for providing the explicit dependency on the properties of the factor graph and helps us to explicitly show the dependency on the number of possible labels, and our theoretical analysis reveals that structured prediction can be generalized well even if it has extremely large size of the output space and under what conditions can there be a tighter generalization bound.
>
> Thank you very much for this review.
> We will add a more detailed discussion of [1], [2], and [3] in the new version.
>
> [1] B. London, B. Huang, and L. Getoor. Stability and generalization in structured prediction. The Journal of
> Machine Learning Research, 17(1):7808–7859, 2016.
>
> [2] B. London, B. Huang, B. Taskar, and L. Getoor. Collective stability in structured prediction:
> Generalization from one example. In International Conference on Machine Learning,
> 2013.
>
> [3] B. London, B. Huang, B. Taskar, and L. Getoor. PAC-Bayesian collective stability. In
> Artificial Intelligence and Statistics, 2014.
>
> **R: I was not able to see a discussion on the limitations of their analysis.**
>
> **A**: Regarding the sample size $n$, the sharpest generalization bound provided by this paper is limited to $1 / n$. As claimed in Section 4, how to obtain a super-linear learning rate in high probability is a problem that we are very concerned about.

---

> ### Author Response · Authors · 2021-09-01
> **Response**
>
> Dear Reviewer rCxV:
>
> Thank you very much for your recognition of the technical contributions of this paper. We promise that we will definitely invite native English speakers to carefully proofread the typographical and grammatical errors in this paper line by line.

---

### Official Review · Reviewer_i4W4 · 2021-07-19

**Rating:** 6
**Confidence:** 4

**Summary:**

The authors consider the problem of bounding the error between the true risk and the empirical risk for structured prediction with a large number of possible labels. By adding fairly weak assumptions that are satisfied for common cases, they improve the dependence on the number of labels $d$ from $O(\sqrt{d})$ to $O(\log d)$ and/or the rate of convergence from $1/\sqrt{n}$ to $1/n$. They then analyze the generalization risk when there may be an infinite number of labels but the family of scoring functions has small complexity as defined by covering number.

**Limitations And Societal Impact:**

No issues.

**Main Review:**

Overall I found the paper to be fairly clear and theoretically interesting. The authors improve the best known generalization results for important cases of structured learning. They show faster convergence of the theoretical loss functions and better dependence on the number of labels. Additionally, the proofs of these results seem theoretically interesting.

The paper can do a better job connecting to practice by proving examples where the framework holds and when there is a large number of labels. Currently the paper is purely theoretical and abstract. The authors can improve the paper by
(1) describing practical situations where there is a large number (or even infinite) labels.
(2) describe practical or theoretically relevant situations where the covering number assumptions holds.

The section with covering numbers is a bit unclear. An example problem where the covering number assumption holds can improve the clarity of the assumption. Additionally a practical problem where the assumption holds and/or there are an infinite number of labels can improve the impact of these results.

Overall I lean towards accept due to the theoretical results, however the paper can be improved by providing more insight into how the results (specifically the improvement in convergence for large/infinite label situations and the results that assume reasonable covering number) can improve the impact of the paper.

**Time Spent Reviewing:**

3

---

> ### Author Response · Authors · 2021-08-10
> **Response to Reviewer i4W4**
>
> **We sincerely appreciate your invaluable and constructive comments.**
>
> **R: Currently the paper is purely theoretical and abstract. The authors can improve the paper by (1) describing practical situations where there is a large number (or even infinite) labels. (2) describe practical or theoretically relevant situations where the covering number assumptions holds.**
>
> **A**: (1.) Example 1: Consider the pairwise Markov networks with a fixed number of substructures $l$ studied by [1], the corresponding factor graph in our paper has $l$ nodes, $|H|\_i = l$, and the maximum size of $\mathcal{Y}\_h$ is $d_i = c^2$ if each substructure of a pair can be assigned one of $c$ class. $c$ may be extremely large in some practical applications, for instance, in part-of-speech tagging. We further consider the Hamming loss $L(y,y') = \frac{1}{l} \sum_{k = 1}^{l} \mathbb{I}_{y_k \neq y'_k}$ as in [1]. If we apply Corollary 1 to the pairwise Markov network and divide the bound through by $l$ to normalize the loss as in [1], we obtain a generalization bound of $\mathcal{O}\left( \frac{  \ln n }{\rho \sqrt{n}}   \left( \log^{\frac{1}{2}} (n^2 ) + \log^{\frac{1}{2}} (lc) \right ) + \sqrt{\frac{\log 1/\delta}{n}} \right)$, which has a lower order dependency of the output space size.
>
> Example 2: Consider the special case of structured prediction: multi-class classification, we have $|H|_i = 1$ and $d_i = c$, where $c$ is the number of classes. $c$ may be extremely large since many challenging applications, such as photo and video annotation and web page categorization,
> can involve tens or hundreds of thousands of
> classes [2]. For instance, practical web page categorization datasets *AmazonTitles-3M* and *Amazon-3M* have 2,812,281 labels, which are much larger than the training samples, please refer to [3].
>
> (2.) Our covering number conditions are assumed on function class $\mathcal{F}_h$. We take the linear function class and the kernel function class for instance to show the two covering number assumptions are easy to be satisfied.
> Theorem 4 in [4] and Corollary 9 in [5] give the covering number bound of linear function class, which satisfies Assumption 3 with $p=2$. Lemma 19 in [6] extends Theorem 4 in [4] to reproducing kernel Hilbert space, which satisfies Assumption 3 with $p=2$. If the linear function class is specified by the commonly used Gaussian Kernel, the covering number bounds in Theorem 15 of [6] and Theorem 5 of [7] satisfy Assumption 3 with $0<p\leq 1$. For more covering number bounds of linear function classes, please refer to [8]. Application 4 in our main paper provides a covering number bound satisfying Assumption 4 if the parameter $w$ is restricted to a euclidean ball.
>
> Thank you very much for this review. We will add a more detailed discussion in the new version.
>
> [1] B. Taskar, C. Guestrin, and D. Koller. Max-margin markov networks. Advances in neural information processing systems, 16:25, 2004.
>
> [2] M. Varma and J. Langford. (2013). NIPS Workshop on extreme Classification. [Online].
>
> [3] Bhatia, K. and Dahiya, K. and Jain, H. and Kar, P. and Mittal, A. and Prabhu, Y. and Varma, M. The extreme classification repository: Multi-label datasets and code. [Online].
>
> [4] T. Zhang. Covering number bounds of certain regularized linear function classes. Journal of Machine Learning Research, 2(Mar):527–550, 2002.
>
> [5] Kakade, S. M., Sridharan, K., and Tewari, A. On the Complexity of Linear Prediction: Risk Bounds, Margin Bounds, and Regularization. In Advances in Neural Information Processing Systems, 2008.
>
> [6] K. Musayeva. Generalization Performance of Margin Multi-category Classifiers. PhD thesis, Université de Lorraine, 2019.
>
> [7] F. Muhammad and I. Steinwart. Learning Rates for Kernel-Based Expectile Regression. Machine Learning, vol. 108, no. 2, 2019, pp. 203–227.
>
> [8] Williamson, Robert C., et al. Entropy Numbers of Linear Function Classes. In Conference on Computational Learning Theory (COLT), 2000.

---

> > ### Comment · Reviewer_i4W4 · 2021-08-31
> > **Response**
> >
> > Thank you for your response. Please include this discussion in the next version of the paper. I especially like the discussion on how the bound improves on real datasets in [3].

---

> > > ### Author Response · Authors · 2021-09-01
> > > **Response**
> > >
> > > Dear Reviewer i4W4:
> > >
> > > Thank you very much for your recognition of this paper and this discussion. We will definitely add this discussion in the next version of the paper.

---

### Official Review · Reviewer_44Ms · 2021-07-31

**Rating:** 6
**Confidence:** 4

**Summary:**

The paper provides generalization bounds for structured prediction learning with a more general setting where each input is associated with its own scoring function. The scoring functions are represented using factor graphs. The authors provide three main results and improve the dependency of the generalization error either on the number of samples or factor and label size. Three classes are considered smooth loss, Lipschitz loss and functions with controlled capacity (stated in terms of their covering number).

**Main Review:**

The main contribution of the paper is a unified generalization bound that has simultaneously better dependency on the cardinality of possible labels and the sample size n.

The results are all solid, and the proofs seem to be fine with when I checked them in passing. The proofs are all based on Rademacher complexity analysis and local Rademacher complexity analysis and bounding them using covering numbers by Dudley’s inequality and generic chaining.

Overall, the results of the paper are interesting, but the writing and presentation can be significantly improved. Besides, I am interested in author’s answer to my comment about the reference [6].

- The paper is hard to read. It has many typos and unpolished statements. For the main paper, the authors could dispense with many terms and keep it to a minimum for better readability.

- The paper [6] provides faster that 1/n convergence, but this is not discussed in the main paper.  The idea of factorization is also present there. It is important to compare this reference with the results of the paper, since it potentially provides better sample complexity.

- The dependency on the sample size in the current analysis is limited to 1/n or 1/\sqrt{n}. Is this an artifact of using Rademacher complexity analysis? I wonder if Rademacher complexity analysis permits getting better convergence than this.


**Time Spent Reviewing:**

3

---

> ### Author Response · Authors · 2021-08-10
> **Response to Reviewer 44Ms**
>
> **We sincerely appreciate your invaluable and constructive comments.**
>
> **R1: The paper is hard to read. It has many typos and unpolished statements. For the main paper, the authors could dispense with many terms and keep it to a minimum for better readability.**
>
> **A**: We will carefully proofread the paper to eliminate all typos, and we promise that we will have this work proofread by native English speakers to identify the typographical and grammatical errors. We will make a major revision to improve our presentations for better readability.
>
> **R2: The paper [6] provides faster that 1/n convergence, but this is not discussed in the main paper. The idea of factorization is also present there. It is important to compare this reference with the results of the paper, since it potentially provides better sample complexity.**
>
> **A**:
> [6] considers the framework from [1], which introduced the implicit embedding approach, and leverages the fact that learning a discrete output space is easier than learning a continuous counterpart, deriving refined calibration inequalities. Then, [6] uses exponential concentration inequalities to turn the refined results into fast rates under the generalized Massart's or Tsybakov margin condition. Different from [6], our generalization bounds are built upon the factor graph decomposition approach. The research ideas and proof techniques of the two papers are very different. Please permit us to elaborate on the advantages of our results over [6].
>
> 1: Regarding sample size $n$, Theorem 6 in [6] shows an excess risk bound of $\mathcal{O}(n^{-\frac{1+\alpha}{2}})$, where $\alpha > 0$ and is a parameter of the generalized Tsybakov margin condition, characterizing the hardness of the discrete problem. Although [6] proposed this interesting and important generalization bound, the bound is presented in expectation. Compared with the result of [6], our result is presented in high probability, which is beneficial to understand the generalization performance of the learned model when restricted to samples as compared to the rates in expectation.
>
> 2: Generalization error bounds studied in [6] require stronger assumptions. For instance, Theorem 6 is proposed under Bilinear loss decomposition condition, exponential concentration inequality condition, generalized Tsybakov margin condition, bounded loss, and finite prediction space condition. Compared with [6], our bounds are presented without the margin conditions. Recently, there is also some work devoted to providing fast classification rates without standard margin conditions in binary classification, for instance [2]. Moreover, our Lipschitz continuous condition and smoothness condition is assumed on the margin function. These assumptions and the capacity condition are pretty milder and easily to be satisfied, as discussed in the main paper.
>
> 3: Factor graph decomposition is beneficial for providing the explicit dependency on the properties of the factor graph and helps us to explicitly show the dependency on the number of possible labels, which sheds new light on the role played by the graph in generalization. In our theoretical analysis, we not only improve the dependence on the sample size $n$ but also improve the dependence on the size of the output space, explaining that structured prediction can still generalize well in the extremely large output labels case.
>
> Thank you very much for this review. We will add a more detailed discussion to the new version.
>
> [1] C. Ciliberto, L. Rosasco, and A. Rudi. A general framework for consistent structured prediction with implicit loss embeddings. Journal of Machine Learning Research, 21(98):1–67, 2020.
>
> [2] B. Olivier, and N. Zhivotovskiy. Fast Classification Rates without Standard Margin Assumptions. Information and Inference: A Journal of the IMA, 2021.
>
> [6] V. Cabannes, A. Rudi, and F. Bach. Fast rates for structured prediction. In Conference on Learning Theory, 2021.
>
> **R3: The dependency on the sample size in the current analysis is limited to $1/n$ or $1/\sqrt{n}$. Is this an artifact of using Rademacher complexity analysis? I wonder if Rademacher complexity analysis permits getting better convergence than this.**
>
> **A**: Mostly yes. Rademacher complexity analysis often provides a learning rate of $\mathcal{O}(1/n)$ or $\mathcal{O}(1/\sqrt{n})$. But there are also some works on stochastic optimization that provide the faster convergence rate of $\mathcal{O}(1/n^2)$ [3, 4]. Their results require an assumption that the optimal expected risk is sufficiently small. Therefore, Rademacher complexity analysis allows obtaining better convergence. We are also very concerned about how to use Rademacher complexity or other tools to obtain faster convergence rates in high probability for structured prediction learning. This is a problem we want to study in the future, as discussed in Section 4.
>
> [3] M. Liu, X. Zhang, L. Zhang, R. Jin, and T. Yang. Fast rates of
> erm and stochastic approximation: Adaptive to error bound conditions. In Advances in
> Neural Information Processing Systems, 2018.
>
> [4] L. Zhang, T. Yang, and R. Jin. Empirical risk minimization for stochastic convex optimization: $\mathcal{O}(1/n)$-and $\mathcal{O}(1/n^2
> )$-type of risk bounds. In Conference on Learning
> Theory, 2017.

---

### Comment · Area_Chair_j4WL · 2021-08-20
**Relationship to other bounds with fast rates**

I am hoping the authors can discuss the relationship between their work and some other risk bounds that achieve the fast rate.

1. Any PAC-Bayesian bound of the "little kl" form can be converted to a bound of the form $R(Q) - \hat{R}(Q) \leq \tilde{O}\big( \sqrt{\frac{\hat{R}(Q) KL(Q||P)}{n}} + \frac{KL(Q||P)}{n} \big)$ using a technique due to McAllester (COLT, 2003). When the empirical risk is small, the bound achieves the fast $\tilde{O}(1/n)$ rate (assuming the KL complexity term is controlled).
2. The empirical Bernstein bounds of Maurer & Pontil (COLT, 2009) are of a form $R(f) - \hat{R}(f) \leq \tilde{O}\big( \sqrt{\frac{Var}{n}} + \frac{1}{n} \big)$ (ignoring the covering number terms). When the variance of the loss is small, the bound is $\tilde{O}(1/n)$.

Granted, these papers do not deal with structured prediction, but it seems to me that any structured prediction risk bound that uses PAC-Bayes or covering numbers could be adapted for these techniques. I would like the authors to comment on this (and add this discussion to the paper).

### References
* D. McAllester. "Simplified PAC-Bayesian Margin Bounds." COLT, 2003.
* A. Maurer and M. Pontil. "Empirical Bernstein Bounds and Sample Variance Penalization." COLT, 2009.

---

> ### Author Response · Authors · 2021-08-24
> **Response to Area Chair j4WL**
>
> **We sincerely appreciate your invaluable and constructive comments.**
>
> **1:** In [1], David McAllester presents the following PAC-Bayesian theorem: $R(Q) - \\hat{R}(Q) \\leq \\widetilde{\\mathcal{O}}\\left(\\sqrt{\\frac{\\hat{R}(Q)KL(Q\\|P)}{n}} +\\frac{KL(Q\\|P)}{n} \\right)$. As Area Chair j4WL claimed, if the empirical risk $\hat{R}(Q)$ is small compared to $\frac{KL(Q\|P)}{n}$, then the last term dominates and the generalization bound scale as $\widetilde{\mathcal{O}}\left( \frac{1}{n} \right)$ when the KL complexity term is also small. Recently, similar fast rate PAC-Bayesian theorems are also presented in Theorem 1.2.6 in [2], Theorem 2.2 in [3], and references therein. Although the fast rate PAC-Bayes has already been established, however, to our best knowledge, how to use the PAC-Bayesian framework to establish fast rates for structured prediction is unexplored. To be specific, David McAllester uses the PAC-Bayesian theorem to provide margin guarantees for the simplest classification problem, binary classification [1]. Subsequently, David McAllester provides generalization bounds for structured prediction tasks involving language models [4], also using the above-mentioned PAC-Bayesian theorem in [1]. [5-7] further extend [4] to more complex learning scenarios of structured prediction: [5] extends [4] to the random structured outputs of maximum loss; [6] extends [4] to Maximum-A-Posteriori perturbation models; [7] extends [4] by including latent variables. However, the generalization bounds in Corollary 1 and Theorem 3 in [1] are of slow order $\widetilde{\mathcal{O}}\left( \frac{1}{\sqrt{n}} \right)$, and as we claimed in Section 1 in the main paper, regarding the sample size $n$, statistical errors in [4-7] are also all stated in slow order $\widetilde{\mathcal{O}}\left( \frac{1}{\sqrt{n}} \right)$. One reason is that Rademacher complexity is often exploited in these papers, leaving a term of slow order $\widetilde{\mathcal{O}}\left( \frac{1}{\sqrt{n}} \right)$. For instance, Rademacher complexity is used to bound a stochastic quantity in the proof of Theorem 2 in [5], to bound a statistical error in the proof of Theorem 1 in [6], and in the proof of Theorem 3 in [7]. Therefore, we think that combining the fast rate PAC-Bayesian theorem and the sharper Rademacher bounds obtained in our paper may be an approach to derive tighter PAC-Bayesian bounds for structured prediction. But this is just a preliminary idea. We thank Area Chair j4WL for proposing this important question. We deem that obtaining sharper PAC-Bayesian guarantees is a very interesting and important problem since a salient advantage of PAC-Bayes is that this theory often ships with powerful
> generalization bounds, of which many are minimax optimal and show the unparalleled performance of the corresponding learning algorithm for specific tasks [8]. We will add a more detailed discussion in the new version.
>
> [1] D. McAllester. "Simplified PAC-Bayesian Margin Bounds." COLT, 2003.
>
> [2] O. Catoni. "PAC-Bayesian Supervised Classification: The Thermodynamics of Statistical
> Learning." Vol. 56. Lecture Notes – Monograph Series. Institute of Mathematical Statistics,
> 2007.
>
> [3] J. Yang, S. Sun, and D.M. Roy. "Fast-Rate PAC-Bayes Generalization Bounds via Shifted Rademacher Processes." NeurIPS, 2019.
>
> [4] D. McAllester. "Generalization bounds and consistency." In Predicting structured data, 2007.
>
> [5] J. Honorio and T. Jaakkola. "Structured prediction: from gaussian perturbations to linear-time principled algorithms." UAI, 2016.
>
> [6] A. Ghoshal and J. Honorio. "Learning maximum-a-posteriori perturbation models for structured prediction in polynomial time." ICML, 2018.
>
> [7] K. Bello and J. Honorio. "Learning latent variable structured prediction models with gaussian perturbations." NeurIPS, 2018.
>
> [8] G. Benjamin. "A Primer on PAC-Bayesian Learning." ICML, 2019.
>
>
> **2:** Theorem 6 in [9] shows the following generalization bound: $R(f) - \\hat{R}(f) \leq \\widetilde{\\mathcal{O}} \\left(\\sqrt{\\frac{Var\_n(f,S) \\ln (\\mathcal{N}\_{\\infty}(1/n,\\mathcal{F},2n) / \\delta)}{n}} + \\frac{\\ln (\\mathcal{N}\_{\\infty}(1/n,\\mathcal{F},2n) / \\delta)}{n} \\right)$, where $\mathcal{F}$ is the loss function class. As Area Chair j4WL claimed, if the variance of the loss and the covering number of the function class $\mathcal{F}$ are small, this generalization bound scale as $\widetilde{\mathcal{O}}\left( \frac{1}{n} \right)$. To explore different learning rates of structured prediction under different conditions, for instance, smoothness curvature and space capacity conditions, instead of assuming directly that the variance of the loss is small, we exploit Theorem 2.1 in [10] and the property of sub-root functions to transform an upper bound with the variance to bound with a fixed point of the local Rademacher complexity, please refer to the proof of Proposition 2. Moreover, assuming directly the covering number on the function class $\mathcal{F}$ will ignore the factor graph property of structured prediction since the factor graph is reflected in the scoring function, not the loss function. In the proof involving covering numbers, we exploit the covering number to bound the local Rademacher complexity and construct relationships of covering numbers among different function classes (please refer to the proofs of Proposition 1 and Theorem 3 in Appendix), which thus permit us to show the explicit dependency on the properties of the factor graph and the dependency on the number of possible labels.
>
> Therefore, our proof and Theorem 6 in [9] all require the variance and the covering number to be small to obtain sharper generalization bounds. However, for the complex structured prediction problem, it requires fined analysis. We thus exploit different implementation modes. Thank you very much for this review, we will add a more detailed discussion in the new version.
>
> [9] A. Maurer and M. Pontil. "Empirical Bernstein Bounds and Sample Variance Penalization." COLT, 2009.
>
> [10] P. L. Bartlett, O. Bousquet, S. Mendelson, et al. Local rademacher complexities. The Annals of Statistics, 33(4):1497–1537, 2005.

---

> > ### Comment · Area_Chair_j4WL · 2021-09-01
> > **Thanks for the explanation**
> >
> > Thank you for this explanation. I apologize for this delayed response.
> >
> > Re: PAC-Bayesian bounds for structured prediction, the situation seems to be as follows: While there are PAC-Bayes bounds [1] that obtain a fast rate (for small empirical risk), all PAC-Bayes applications for structured prediction have used a looser form of the PAC-Bayes theorem, which is dominated by $\tilde{O}(1/\sqrt{n})$. If these works had instead started from the tighter form, they would inherit the fast rate (when the empirical risk is small). Thus, the authors argue that no PAC-Bayesian bounds for structured prediction obtain the fast rate. Fair enough.
> >
> > I guess my next question would be _why_? Why has no one used a tighter PAC-Bayes bound for structured prediction? (Note: I don't expect the authors to answer this.)
> >
> > One reason could be that McAllester's bound is too restrictive for structured prediction. It starts from the kl-type bound, where the (emp.) risk is stated as a KL divergence between Bernoulli distributions. It just occurred to me that this definition might not handle common structured prediction losses, such as the Hamming loss. I think McAllester might have once argued (in a talk I saw) that the bound can actually accommodate _any_ $[0,1]$-bounded loss... But since I am uncertain about this, I won't hold this one critique against the paper.
> >
> > I do hope that the authors add some discussion of McAllester's fast-rate bound, and other structured prediction PAC-Bayes bounds, to the paper (if there's room!).

---

> > > ### Author Response · Authors · 2021-09-02
> > > **Thanks for the response**
> > >
> > > Dear Area Chair j4WL:
> > >
> > > As you claimed, instead of using the relative deviation PAC-Bayesian bound, McAllester used a looser form for structured prediction. The bounds presented by McAllester are thus of order $\widetilde{\mathcal{O}}\left( \frac{1}{\sqrt{n}} \right)$ [4]. [5-7] extend [4] to more complex learning scenarios of structured prediction. Under the influence of many factors, their bounds are also of slow order, for instance, the looser form of McAllester's bound and the global Rademacher complexity. Regarding why has no one used a tighter PAC-Bayes bound for structured prediction, we think this is a very interesting question. Moreover, we think that you may have provided an important idea to explain this. We will combine your thoughts with ours to make a detailed discussion on McAllester's fast-rate bound and other structured prediction PAC-Bayes bounds in the new version of this paper.

---

### Author Response · Authors · 2021-08-29
**New Response to AC and All Reviewers**

Dear AC and Reviewers,

Thank you very much again for the great efforts and the valuable comments. We have responded carefully and in detail to the main concerns. We hope you will be satisfied with these responses. As the discussion phase is about to close, we are very much looking forward to hearing from you about any further feedback. We will be very happy to clarify any further concerns (if any).

Finally, we would like to emphasize that we will do our best to proofread all typographical and grammatical errors.

Best,

Authors

---

### Decision · Program_Chairs · 2021-09-27

**Decision:**

Accept (Poster)

**Comment:**

This paper proves generalization bounds for structured prediction that are sharper than existing bounds. In particular, under certain conditions, they obtain the "fast rate" of $\tilde{O}(1/n)$. Also of interest is the fact that the bounds are logarithmically dependent on the cardinality of each output variable; meaning, the bounds accommodate very large label sets -- which is the case for things like sentence completion or object classification. The key technical innovation that enables the tighter bounds seems to be a refined analysis of the factor graph representation.

The reviews were mixed. The biggest complaint (shared by multiple reviewers) is that the paper is difficult to read. Having only skimmed it myself, I agree that there are a bunch of grammatical/syntactical errors that should have been caught before submission, but in this case I don't think it's enough (on its own) to warrant rejection; the paper is readable. These types of mistakes can be fixed in a camera-ready after a detailed proofreading. **I _strongly_ suggest the authors go over the paper "with a fine-toothed comb" (or ask someone else to).**

Other than the writing, the presentation seems pretty good. I think it was wise of the authors to present the main results in asymptotic notation, to make them easier for the reader to digest. Similarly, it was wise to defer proofs (which are quite involved) to the appendix and give just proof sketches. That said, Reviewer Mdym makes a good point that "it's hard to pinpoint which technical contributions of the work are instrumental to the derivation of the results." The authors responded, and I hope their explanation makes it into the paper.

Another complaint is that the paper ignores some related works (London et al., JMLR 2016; Maurer & Pontil, COLT 2009; McAllester, COLT 2013). The authors responded to these suggestions, providing convincing comparisons, during the discussion period. I ask them to please add this discussion to the paper.

As for the content, the results are solid. I'm glad that progress is being made on generalization in structured prediction. It's nice that the paper gives bounds for both the margin-rescaling (additive) and slack-rescaling (multiplicative) form of the max-margin loss.

It's worth noting that the paper is entirely theoretical, with no experiments. While this situation is not unusual, it's becoming more common these days for theory papers to include a numerical study that tests the assumptions or evaluates the bounds on real data, to see if the bounds are still meaningful. Since the paper's primary claim is that these bounds are tighter than others, it would be interesting to see if the claim holds in practice. Are these bounds meaningful when others are vacuous? So, while I think it's OK to not provide experiments, I think the paper would have been stronger with experiments.

I am recommending Accept because the work is interesting and valuable, and I believe the problems (writing and related work) can be worked through for the camera-ready.